# Point-Identification of a Robust Predictor Under Latent Shift with Imperfect Proxies

**Zahra Rahiminasab**                                                     *zahra.rahiminasab@aalto.fi*
*Department of Computer Science*
*Aalto University*

**Reza Soumi**                                                                   *reza.soumi@aalto.fi*
*Department of Computer Science*
*Aalto University*

**Arto Klami**                                                                   *arto.klami@helsinki.fi*
*Department of Computer Science*
*University of Helsinki*

**Samuel Kaski**                                                             *samuel.kaski@aalto.fi*
*ELLIS Institute Finland*
*Department of Computer Science*
*Aalto University*
*Department of Computer Science*
*Manchester University*

**Reviewed on OpenReview:** *https://openreview.net/forum?id=QFJuVreJDC*

## Abstract

Addressing the domain adaptation problem becomes more challenging when distribution shifts across domains stem from latent confounders that affect both covariates and outcomes. Existing proxy-based approaches that address latent shift rely on a strong completeness assumption to uniquely determine (point-identify) a robust predictor. Completeness requires that proxies have sufficient information about variations in latent confounders. For imperfect proxies the mapping from confounders to the space of proxy distributions is non-injective, and multiple latent confounder values can generate the same proxy distribution. This breaks the completeness assumption and observed data are consistent with multiple potential predictors (set-identified). To address this, we introduce latent equivalent classes (LECs). LECs are defined as groups of latent confounders that induce the same conditional proxy distribution. We show that point-identification for the robust predictor remains achievable as long as multiple domains differ sufficiently in how they mix proxy-induced LECs to form the robust predictor. This domain diversity condition is formalized as a cross-domain rank condition on the mixture weights, which is substantially weaker assumption than completeness. We introduce the Proximal Quasi-Bayesian Active learning (PQAL) framework, which actively queries a small, targeted set of diverse domains that satisfy this rank condition. PQAL can recover the point-identified predictor, demonstrates robustness to varying degrees of shift and outperforms previous methods on synthetic data and semi-synthetic dSprites, IHDP, ACS Folktables datasets.

## 1 Introduction

Domain adaptation transfers models trained on labeled source domains to a target domain, where labeled data is either unavailable entirely (Tsai et al., 2024) or restricted to a few samples (few-shot domain adaptation) (Motiian et al., 2017). To successfully adapt a trained predictor from the source to the target domain,

some assumptions about the type of data distribution shift must be made (David et al., 2010). Traditionally, covariate (Shimodaira, 2000) or label (Lipton et al., 2018) shift assumptions are used. However, these assumptions do not always hold in real-world settings. In such settings, let $S$ source domains and a target domain are indexed by $Z \in \mathcal{Z} = \{1, \ldots, S, S+1\}$. Then, the shift in distribution between source and target domains is due to a latent confounder $U \in \mathcal{U}$, such that $P(U|Z=i) \neq P(U|Z=S+1)$ for all $i \in \{1, \ldots, S\}$, which affects the covariates $X \in \mathcal{X} \subseteq \mathbb{R}^{d_X}$ and the outcomes $Y \in \mathcal{Y} \subseteq \mathbb{R}^{d_Y}$ (Alabdulmohsin et al., 2023; Tsai et al., 2024).

Due to such unobserved confounders $U$, the predictors may learn spurious correlations between $X$ and $Y$. Previous studies, such as Tsai et al. (2024); Alabdulmohsin et al. (2023); Prashant et al. (2025), use proxies $W \in \mathcal{W}$ to address the latent shift. They assume that the proxy is informative enough to uniquely identify (point-identify) the parameter or function of interest, based on the probability of the observed data (Manski, 2005). For discrete unobserved confounders and proxies, being informative translates to the invertibility of the conditional distribution of the proxy given the unobserved confounder $P(W|U)$, which is often formalized as a rank condition (Alabdulmohsin et al., 2023; Prashant et al., 2025; Tsai et al., 2024). In contrast, when unobserved confounders and proxies are continuous (Tsai et al., 2024), the completeness assumption must hold, which ensures injectivity of the conditional operator induced by $P(W|U)$(Miao et al., 2018). These assumptions break in practice when proxies are imperfect or insufficiently informative. More precisely, when the $P(W|U)$ is not invertible, or the conditional expectation operator is not injective. In such cases, parameters or functions of interest can only be partially identified (set-identified).

In human-in-the-loop learning, proxies are not passively observed; instead, they are actively collected through budget-limited interactions with a human expert. As a result, these proxies are often noisy or weak (Sloman et al.). Examples include expert annotations (Srivastava et al., 2020) or feedback (Nikitin & Kaski, 2022). Since acquiring proxies requires costly interactions, only a limited number of proxies can be obtained in practice. Consequently, it is important to identify robust predictors while strategically selecting these imperfect proxy queries under strict budget constraints. We address the aforementioned problems by making the following contributions.

- **Relaxing completeness via latent equivalent classes (LECs):** This paper addresses domain adaptation under latent shift, when proxies are imperfect and do not satisfy the standard completeness assumption. When the completeness assumption does not hold, distinct values of the latent confounder may generate the same conditional proxy distribution $P(W|U)$. We introduce latent equivalent classes (LECs) as sets of latent confounder values that are observationally indistinguishable given the proxy. Although individual latent confounder values cannot be recovered, the conditional proxy distribution in each domain can be represented as a mixture over these LECs. This decomposition enables tracking latent shifts by leveraging variation in mixture weights across domains.

- **Point-identification via distinguishing environment set:** Based on the obtained mixture formulation, we prove that the robust predictor ($\mathbb{E}[Y|X=x, Z=S+1]$, where $X$ is a covariate and $Y$ is an outcome. In addition, environments are indexed by discrete variable $Z$.) is point-identified as long as the observed domains exhibit sufficient diversity in their mixture weights over LECs. We call such environments distinguishing environments $Z^\star \subseteq \mathcal{Z}$. This condition is formalized as a simple cross-domain rank condition. We formally prove that this rank condition is strictly weaker than the completeness assumption, as it depends on variation across environments rather than on the injectivity of the conditional operator in each environment.

- **Proximal Quasi-Bayesian Active Learning (PQAL):** We propose the Proximal Quasi-Bayesian Active Learning (PQAL) framework to train a point-identified robust predictor. PQAL actively queries proxies to satisfy the required rank condition. We show that PQAL recovers a point-identified predictor and achieves lower mean squared error than state-of-the-art methods across varying degrees of latent distribution shift on synthetic and semi-synthetic dSprites, IHDP, ACS Folktables datasets.

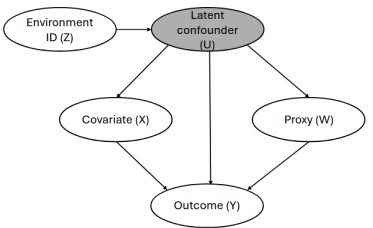

Figure 1: Causal diagram that complies with the latent shift assumption

## 2 Problem formulation

Let $\mathcal{X}$ denotes the space of observed covariates $X$, and $\mathcal{Y}$ the space of outcomes $Y$. We define a set of $S$ sources and a target environment with domain IDs $Z \in \{1, \ldots, S, S+1\}$. In every environment, both the covariates and the outcomes are influenced by an unobserved confounder $U$, which varies across the environments. That is, $p(U|Z = z)$. This distribution is also unknown, but we assume that partial knowledge about it can be obtained via a proxy $W$ that contains indirect and imperfect information regarding changes in the distribution of unobserved confounders $U$. When proxies are imperfect, multiple latent values from $U$ can induce the same conditional proxy distribution $P(W \mid U)$. For instance, periodically repeating confounders, like orientation where $2\pi + \theta$ and $\theta$ induce the same proxy distribution (see dSprites dataset in the Section 6.1 for an example, where we consider domains that differ by such orientation).

**Assumption 1** (Latent shift (from Tsai et al. (2024))). *Latent shift is translated to the following conditions:*

1. *The shift between source distribution $P$ and a target distribution $Q$ stems from unobserved confounder $U$. Therefore $P(U) \neq Q(U)$. Also, $\forall z_i, z_j$ when $i \neq j$, $P(U|z_i) \neq P(U|z_j) \neq Q(U)$.*

2. *Invariant distributions: a conditional distribution of observed covariates, proxy, and an outcome for an unobserved confounder remain invariant between source and target domains $P(R|U) = Q(R|U)$ for $R \subseteq \{X, Y, W\}$.*

Figure 1 illustrates the causal DAG that complies with the latent shift (Assumptions 1).

**Problem:** Given $m$ labeled data from $S$ source domains with observed variables $\{(x_i, w_i, z_i, y_i)\}_{i \in \{1, \ldots, m\}}$, where $W$ is imperfect proxy for the latent confounder $U$. We observe variables $\{(x_j, z_j)\}_{j \in \{1, \ldots, n\}}$ and we may also observe $w_j$ from target domain $(S+1)$. The goal is to uniquely determine (Point-identify) the optimal predictor $\mathbb{E}[Y|X = x, Z = S+1]$ on a target domain based on the joint distribution of the observed data.

**Learning setup:** Proxies are usually obtained through costly interactions, such as expert feedback. Therefore, only a limited number of proxies can be queried from a pool of candidate samples. This practical constraint motivates the use of an active selection strategy to query the most informative proxies for learning a point-identified robust predictor.

---

**Motivating Example: Robust Drone Design Under Manufacturing shift**

**Setting:** Consider $S$ different manufacturers (indexed by $Z$) constructing drones characterized by covariates $X$ (e.g., battery mass, wing chord). Both the covariates $X$ and the function that maps them to observable properties $Y$ (e.g. flight speed and hovering time) depend on unobservable confounders $U$, such as manufacturing quality. While $U$ cannot be observed directly, we assume a proxy variable $W$ to provide partial and indirect information about it. For example, a human expert inspecting the drones can estimate the overall quality of all manufacturers, but does this in a crude manner: If they cannot differentiate between the quality of some manufacturers they provide the same assessment for them.
**Goal:** Train a point-identifiable predictor estimating the properties $Y$ for a new manufacturer, based on the $\{X, Y, W\}$ triplets for the $S$ manufacturers and the proxy variable $W$ for the new manufacturer.

---

## 3 Notation and Background

**Notation:** For any variable $V \in \{\mathcal{X}, \mathcal{W}, \mathcal{Z}, \mathcal{Y}\}$, let $k : \mathcal{V} \times \mathcal{V} \to \mathbb{R}$ be a positive semi-definite kernel function, and for every $v \in \mathcal{V}$, feature maps are defined as $\phi(v) = k(v, .)$. The Reproducing Kernel Hilbert Space (RKHS) (Murphy, 2012) is defined on $\mathcal{V}$ corresponding to defined kernel functions. $\mathcal{K}_V = [k(v_i, v_j)]$ are gram matrices and $\mathcal{K}_{VV'} = [k(v_i, v'_j)]$ are cross-gram matrices. Also, $\Phi_V(v') = [k(v_1, v') \dots k(v_{|V|}, v')]$. In addition, $\otimes$, $\bar{\otimes}$, and $\odot$ are the tensor product, column-wise Khatri-Rao product, and Hadamard products, respectively. $\mathcal{D}_{\mathcal{LB}} = \{(x_i, w_i, z_i, y_i)\}_{i=1}^{m_1}$ is a set of labeled samples from the source domains. Also, $\mathcal{D}_{\mathcal{PL}} = \{(x_j, z_j)\}_{j=1}^{n}$ is a pool of candidate samples without proxies. In addition, $\mathcal{D}_{\mathcal{PR}}^r = \{(x_j^{(r)}, w_j^{(r)}, z_j^{(r)})\}_{j=1}^{n_{\mathcal{PR}}}$ is a set of queried samples with proxy at round $r$ from source and target environments. Finally, $\mathcal{D}_{\mathcal{TLB}}^r = \{(x_k^{(r)}, w_k^{(r)}, z_k^{(r)}, y_k^{(r)})\}_{k=1}^{n_{\mathcal{TLB}}}$ is a set of labeled samples from a target environment where $n_{\mathcal{TLB}} \ll n_{\mathcal{PR}}$. Throughout the paper, $\lambda$ with different subscripts denotes regularization hyperparameters.

Following Tsai et al. (2024), we adapt identification techniques from proximal causal inference (PCI) to provide point-identification guarantees for predictors. In addition, we use the expected predictive information gain (EPIG) to design an acquisition function that selects the most informative queries for distinguishing LECs. Therefore, we review PCI and EPIG in this section.

### 3.1 Proximal causal inference

Proximal causal inference (PCI) (Tsai et al., 2024) uses proxies to adjust for the effect of unobserved confounders. PCI solves linear inverse problem $\mathbb{E}[Y - h(X, W)|X = x, Z = z] = 0$, utilizing a conditional expectation operator ($\mathcal{T}$) to map a bridge function $h$ to the domain specific predictor ($\mathbb{E}[Y|X = x, Z = z]$) Wang et al. (2021); Bennett et al. (2023). The bridge function $h$ connects covariates $X$ and proxies $W$ to outcome $Y$ by integrating over the unobserved confounder $U$. Formally, PCI estimates the optimal predictor $\mathbb{E}[Y|X = x, Z = z]$ based on a bridge function $h$ and conditional distribution $p(w|x, z)$ by integrating over unobserved confounders $u$ as

$$\mathbb{E}[Y|X = x, Z = z] = \int_{\mathcal{U}} \int_{\mathcal{W}} h(x, w)p(w|u)p(u|x, z)dwdu = \int_{\mathcal{W}} h(x, w)p(w|x, z)dw = \mathcal{T}h. \tag{1}$$

Establishing the point-identification of a predictor $\mathbb{E}[Y|X = x, Z = S + 1]$ in target environment based on Equation 1 generally involves two steps (Miao et al., 2018; Tsai et al., 2024):

1. Establishing the existence of a bridge function based on solvability and regularity assumptions (e.g., Picard/range conditions and square integrability)

2. Point-identification of optimal predictor based on the existence of a bridge and by assuming completeness.

The completeness assumption ensures the injectivity of the conditional expectation operator $\mathcal{T}$ by enforcing the following conditions (Melnychuk et al., 2024):

1. Given square integrable function $l$, $\mathbb{E}[l(U)|x,z] = 0$ for all $(x,z) \in \mathcal{X} \times \mathcal{Z}$ if and only if $l(U) = 0$ almost surely.

2. Given square integrable function $g$, $\mathbb{E}[g(Z)|x,w] = 0$ for all $(x,w) \in \mathcal{X} \times \mathcal{W}$ if and only if $g(Z) = 0$ almost surely.

In a discrete setting, the first completeness condition corresponds to invertibility of the matrix representation of $P(U \mid X, Z)$, which captures how latent variables vary across covariates and environments. The second completeness condition translates to invertibility of the matrix representation of $P(W|U)$, which determines the amount of information proxy carries about the latent confounder. Our framework addresses the failure of the proxy mechanism when the second completeness condition is violated by introducing a cross-domain rank condition based on variation in latent distributions across environments. As our theoretical framework is formulated on general measurable (Borel) spaces, the same arguments apply when the variables are discrete, with integrals replaced by finite sums and operators represented as matrices.

To solve this inverse problem, the bridge function $h$ and the conditional distribution $p(w|x,z)$ are estimated by Kernel Proxy Variable (KPV) (Mastouri et al., 2021). KPV learns the conditional mean operator $\mathcal{T}$ in the reproducing kernel Hilbert Space (RKHS) via $\mathcal{T} = \phi(X) \otimes \mu_{W|X=x,Z=z}$, where $\mu_{W|X=x,Z=z} = \mathbb{E}[\phi(W)|x,z]$ is conditional mean embedding (CME). Then, by combining learned bridge function and CMEs, the optimal predictor $\mathbb{E}[Y|X=x, Z=z]$ is trained.

### 3.2 Expected predictive information gain (EPIG)

Expected predictive information gain (EPIG) selects candidate samples $x$ that maximize expected reduction in predictive uncertainty at a target input $x_\star \sim p_\star(x_\star)$ (Smith et al., 2023). EPIG is defined based on the conditional mutual information ($I$) function between candidate label $y$ and target label $y_*$ as follows:

$$EPIG(x) = \mathbb{E}_{p_*(x_*)}[I(y; y_*|x, x_*)] \tag{2}$$

## 4 Point-identification of predictor

In this section, we formalize the relationship between an unobserved confounder $U$ and a proxy $W$ by introducing the concept of imperfect proxies. Then, we establish theoretical results for the point-identifiability of $\mathbb{E}[Y|X=x, Z=S+1]$ despite imperfect proxies.

### 4.1 Imperfect proxies and LECs

First, we formalize the relationship between an unobserved confounder $U$ and a proxy $W$ by introducing the concept of imperfect proxies.

**Definition 1** (Imperfect proxies). *The proxy $W$ is imperfect in capturing the latent confounder $U$ if there is more than one value of a latent variable $U$ that can induce its conditional distribution $P(W|U)$. Formally, there exists $u \neq u'$ such that:*

$$P(W \in A|U = u) = P(W \in A|U = u'), \quad \forall A \in \mathcal{A} \quad \text{where } (\mathcal{W}, \mathcal{A}) \text{ is the measurable space of } W \tag{3}$$

Imperfect proxies form groups of latent values that are indistinguishable by proxy $W$. We formalize these groups as latent equivalent classes (LECs).

**Definition 2** (Latent equivalent classes (LECs)). *Two values $u$ and $u'$ of latent variable $U$ belong to the same latent equivalent class $O$ with respect to $W$ ($u \sim_O u'$) if and only if $P(W \in A|U = u) = P(W \in A|U = u')(\forall A \in \mathcal{A})$. Each latent equivalence class $O_k = \{u : u \sim_{O_k} u'\}$ contains all latent confounder values that induce the same conditional proxy distribution. As a result, proxy $W$ cannot distinguish values within*

*the same LEC; however, it can distinguish across different LECs. Therefore, LECs form a partition of the entire unobserved confounder space $\mathcal{U}$. Formally, we have*

$$\mathcal{U} = \bigcup_{j=1}^{|O|} O_j \, and \, O_i \cap O_j = \varnothing \, (\forall i \neq j) \tag{4}$$

The following remark ensures the proxy does not collapse all latent confounder values into a single LEC. Otherwise proxy has no information about domain-induced changes in $U$, and the domain adaptation problem is trivial and unsolvable.

**Remark 1** (Non-degenerate LECs). *To avoid trivial domain adaptation problem, we assume proxy $W$ is non-degenerate, meaning it induces at least two distinct latent classes ($|O| \geq 2$).*

## 4.2 Theoretical results

Lemma 1 shows that the conditional distribution $P(W \mid X = x, Z = z)$ can be decomposed as a mixture over LECs. Establishing this decomposition is a key step for the point-identification of the predictor $\mathbb{E}[Y|X = x, Z = S + 1]$. To ensure the validity of our derivations, the following regularity conditions must hold.

**Assumption 2** (Regularity condition). *We assume the domains $\mathcal{X}$, $\mathcal{Y}$, $\mathcal{W}$, $\mathcal{Z}$, and $\mathcal{U}$ are Borel spaces with $\sigma$-algebra. Also, the conditional distribution $P(W|U)$ exists. In addition, LECs $\{O_j\}_{j=1}^{|O|}$ are measurable sets with respect to the $\sigma$-algebra of $U$.*

**Lemma 1** (Decomposition of Conditional Distribution). *Let $\{O_j\}_{j=1}^{|O|}$ be the latent equivalent classes (LECs) defined in Definition 2. Assume the regularity condition of Assumption 2 holds. For each LEC $O_j$, define its conditional distribution $P_j(A) = P(W \in A \mid U \in O_j)$, the corresponding mixing weight $\pi_j(x, z) := P(U \in O_j \mid X = x, Z = z)$. Under latent shift assumption (Assumption 1), and its corresponding causal DAG (Fig 1), $W \perp (X, Z) \mid U$ holds. This implies that the conditional distribution of $W$, has the mixture representation*

$$P(W \in A|X = x, Z = z) = \sum_{j=1}^{|O|} \pi_j(x, z) P_j(A) \quad \textit{for any measurable set } A \subseteq \mathcal{W}. \tag{5}$$

*Proof.* Refer to Appendix A.1 for the proof. $\qquad\square$

If the mixture weights in Lemma 1 do not vary across environments, then the proxy distribution $P(W|X, Z)$ provides no information to distinguish among different LECs. Therefore, the mixture weights must be linearly independent across environments. We formally define such environments as follows:

**Definition 3** (Distinguishing environment set). *Let $Z^\star = \{z_1, \ldots, z_e\}$ be a set of $e$ source environments. Let $\pi_j(x, z)$ be mixture weights defined in Lemma 1. For almost every $x$ with respect to $P(x)$, we say that $Z^\star$ **distinguishes** latent equivalent classes $\{O_1, \ldots, O_{|O|}\}$, if the matrix $\Pi(x) = [\pi_j(x, z_l)]_{l,j}, l = 1, \ldots, e, j = 1, \ldots, |O|$ has full rank $|O|$.*

With imperfect proxies, the second completeness condition fails, and the operator $g \mapsto \mathbb{E}[g(Z)|x, w]$ is not injective. Nevertheless, Theorem 1 shows that the cross-domain rank condition in Definition 3 is strictly weaker than the first completeness condition over operator $l \mapsto \mathbb{E}[l(U)|x, z]$. With this rank condition, the point-identification of $\mathbb{E}[Y|X = x, Z = S + 1]$ remains possible.

**Theorem 1** (Cross-domain rank condition and completeness). *The cross-domain rank condition in Definition 3 is strictly weaker than the first completeness assumption in the following sense: it only requires injectivity at the level of proxy-induced latent equivalent classes, whereas the first completeness assumption requires injectivity over full latent space.*

*Proof.* Refer to Appendix A.2 for the proof. $\qquad\square$

In practice, the cross-domain rank condition requires non-redundancy of the observed source environments. In other words, these environments must capture fundamentally different distributions of the latent confounder by mixing the proxy-induced LECs in sufficiently diverse ways. Although the mixture matrix $\Pi(x)$ is latent, this requirement is empirically evidenced as a high effective rank of the stacked environment-specific CMEs. We empirically demonstrate this relationship between environment diversity, proxy imperfectness, and effective rank in Section 6.2.

As discussed in Section 3.1, establishing the existence of a bridge is not guaranteed by completeness; it requires specific solvability and regularity assumptions. We formalize the solvability assumptions as follows.

**Assumption 3** (LEC moment condition)**.** *Consider distinguishing environment set $Z^\star$ and mixture weight matrix $\Pi(x) = [\pi_j(x, z_l)]_{l,j}, l = 1, \ldots, e, j = 1, \ldots, |O|$ (Definition 3). Let $r(x, z) = \mathbb{E}[Y \mid X = x, Z = z]$, define $r_x := (r(x, z_l))_{l=1}^e$. Then, we assume the following condition holds:*

1.  ***Compatibility:** For almost surely x, $r_x \in Range(\Pi(x))$. Equivalently, there exists a measurable function $g(x) \in R^{|O|}$ such that $\Pi(X)g(x) = r_x$.*

2.  ***LEC proxy spanning:** $p_j \in L_2(\mathcal{W})$ for all $j = 1, \ldots, |O|$, and the gram matrix $G_{jk} := \int_W p_j(w)p_k(w)dw \quad j, k = 1, \ldots, |O|$ is invertible.*

3.  ***Square-integrability** The function $g(x) = g_1(x), \ldots, g_{|O|}(x)$ can be chosen such that $g_j \in L_2(\mathcal{X})$ for all j.*

**Remark 2** (Role and verifiability of Assumption 3)**.** *Assumption 3 adapts traditional solvability assumptions to the LEC-based framework:*

*   *Compatibility is analogous to Picard/range condition (e.g. Assumption (vii) in Miao et al. (2018); Assumption 2 in Proposition A.2 of Tsai et al. (2024)). It requires that outcome-relevant latent variations are distinguishable across different LECs. Compatibility is not directly testable as LECs are latent. However, it can be tested indirectly. If using more informative proxies or environment diversity raises the effective rank without improving target prediction, this suggests the proxy may not capture the outcome-relevant latent shift.*

*   *LEC proxy spanning corresponds to operator non-degeneracy and boundedness requirements (e.g. Assumption (v) in Miao et al. (2018); Assumption 2 in Proposition A.2 in Tsai et al. (2024). It ensures that proxy distribution induced by different LECs provides non-redundant direction in proxy space. Although the LECs are latent, their mixtures appear in the observable CMEs. Therefore, the singular values and rank of stacked CMEs provide a direct empirical diagnostic for this condition. We explicitly test this diagnostic and show how proxy imperfection impacts the effective rank in section 6.2.*

*   *Square integrability is a standard regularity condition ensuring the bridge lies in a stable $L_2$ function class (e.g. Assumption (vi) from Miao et al. (2018); Assumption 3 in Proposition 4.2 of Tsai et al. (2024)). It is not directly testable for finite data. However, in practice, the use of bounded kernels and ridge regularization restricts the learning process to a stable class, ensuring this condition holds.*

The next theorem shows the existence and set-identifiability of the bridge function, which is required for guaranteeing point-identification of $\mathbb{E}[Y|X = x, Z = S + 1]$.

**Theorem 2** (Existence and set identifiability of bridge function)**.** *Consider imperfect proxies (Definition 1) and let $Z^\star$ be a distinguishing environment set (Definition 3). Also, Lemma 1 holds. Under Assumption 3, the following hold:*

*   *There exists at least one bridge function h such that:*

$$\mathcal{T}_{joint}h = r, \qquad that\ is, \quad \mathcal{T}_z h = r(., z) \quad \forall z \in Z^\star \tag{6}$$

- *The set of all bridge functions solving $\mathcal{T}_{joint}h = r$ is an affine set:*

$$\mathcal{H}(r) = \{h : \mathcal{T}_z h = r(.,z)(\forall z \in Z^\star)\} = h_0 + \bigcap_{z \in Z^\star} ker(\mathcal{T}_z) \tag{7}$$

  *where $ker(\mathcal{T}_z)$ is null space of $\mathcal{T}_z$ and $h_0$ is any particular solution.*

**Proof sketch.** *Under Assumption 3 and Lemma 1 $p(w \mid x, z) = \sum_{j=1}^{|O|} \pi_j(x, z)p_j(w)$, the equation $\mathcal{T}_{joint}h = r$ can be written for each $x$ as the linear system $r_x = \Pi(X)g(x)$ where $g_j(x) = \int_{\mathcal{W}} h(x, w)p_j(w)dw$. The compatibility assumption guarantees such $g(x)$ exists, and the LEC proxy spanning assumption allows constructing $h_0 \in L_2(\mathcal{X} \times \mathcal{W})$ with these moments. Therefore, there exists at least one bridge function that solves $\mathcal{T}_{joint}h_0 = r$. Due to linearity of $\mathcal{T}_{joint}$, any two solutions $h_1$ and $h_2$ to $\mathcal{T}_{joint}h = r$ satisfy $h_2 - h_1 \in ker(\mathcal{T}_{joint}) = \bigcap_{z \in Z^\star} ker(\mathcal{T}_z)$. Therefore, $\mathcal{H}(r) = h_0 + \bigcap_{z \in Z^\star} ker(\mathcal{T}_z)$ (Refer to Appendix A.3 for extended proof).*

Finally, based on the rank conditions for LECs and the existence of a bridge function, we prove point-identification of the predictor $\mathbb{E}[Y|X = x, Z = S + 1]$.

**Theorem 3** (Point-identification of predictor)**.** *Assume conditions of Theorem 2 and Lemma 1 hold. Consider set of distinguishing environments $Z^\star$ (Definition 3). Under these conditions, the predictor $\mathbb{E}[Y|X = x, Z = S + 1]$ is point-identified.*

**Proof sketch.** *By Theorem 2, the set of the bridge functions $\mathcal{H}(r)$ is non-empty. For any two solutions $h_1, h_2 \in \mathcal{H}(r)$, let $\Delta h = h_2 - h_1$, then $\mathcal{T}_{joint}\Delta h = 0$, implying $\mathcal{T}_z \Delta h = 0$ for all distinguishing environment set $z \in Z^\star$. By Lemma 1, $\mathcal{T}_z$ decomposes into latent operators $\mathcal{T}_{O_j}$. By full rank condition of distinguishing environment set, $\mathcal{T}_{S+1}$ can be expressed as linear combination of distinguishing environments operators $\mathcal{T}_{S+1} = \sum_{z \in Z^\star} \gamma_z \mathcal{T}_z$. Consequently, $\mathcal{T}_{S+1}\Delta h = \sum_{z \in Z^\star} \gamma_z(\mathcal{T}_z h) = 0$. This implies $\mathcal{T}_{S+1}h_1 = \mathcal{T}_{S+1}h_2$. Therefore, $\mathbb{E}[Y|X = x, Z = S + 1]$ is point-identified.*

## 5 Proximal Quasi-Bayesian Active Learning (PQAL)

Having established that the predictor is point-identified in the target domain if a cross-domain rank condition holds, we next present a budget-constrained active learning framework for learning such a predictor. The goal is to learn the CME operator and adapt the bridge function to the target domain while considering a limited budget of proxy and label queries from the target. Figure 2 shows an overview of the PQAL framework, including **Initial training:** CMEs and the bridge function are trained by labeled samples from source environments (lines 2-3 of Algorithm 1), **Sample selection:** Most informative samples are selected to query from an external information source (lines 5-15 of Algorithm 1), and **Adaptation:** CMEs in the target domain are trained based on obtained proxies, and source bridge parameters $\alpha_s$ are fine-tuned and updated to $\alpha$ based on samples with both proxy and labels (lines 16-17 of Algorithm 1). The algorithm returns updated bridge parameters $\alpha$ and CMEs.

### 5.1 Initial training

We adapt the KPV-based approach (Tsai et al., 2024; Mastouri et al., 2021) to estimate CMEs and bridge functions. KPV trains CMEs and bridge functions by solving a two-step ridge regression problem. In the first step, given a set of labeled samples from the source domains $\mathcal{D}_{\mathcal{LB}}$, the goal is to solve the optimization problem

$$C_{W|X,Z} = \arg\min_C \frac{1}{m_1} \sum_{i=1}^{m_1} ||\phi(w_i) - C(\phi(x_i) \otimes \phi(z_i))||^2 + \lambda_{CME}||C||^2. \tag{8}$$

Since $\mu_{W|X=x,Z=z} = C_{W|X,Z}(\phi(x) \otimes \phi(z))$, the closed-form of CME is calculated as follows (Tsai et al., 2024; Mastouri et al., 2021):

$$\mu_{W|X=x,Z=z} = \sum_{i=1}^{m_1} b_i(x, z)\phi(w_i), \tag{9}$$

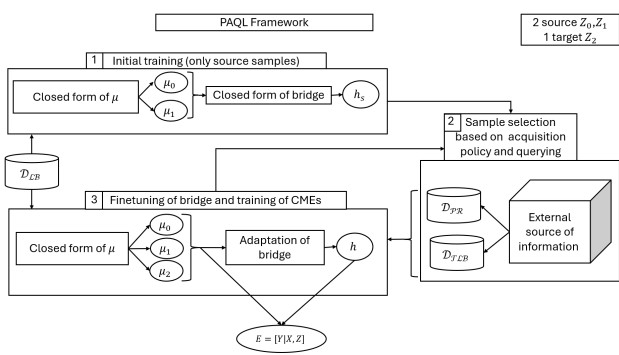

Figure 2: Overview of PQAL framework

where $b(x,z) = (\mathcal{K}_Z \odot \mathcal{K}_X + \lambda_{CME} m_1 I)^{-1} (\Phi_Z(z) \odot \Phi_X(x))$. Then, given set of mutually exclusive labeled samples $\{(\tilde{x}_j, \tilde{w}_j, \tilde{z}_j, \tilde{y}_j)\}_{j=1}^{m_2}$, the bridge function can be estimated by solving the optimization problem (Mastouri et al., 2021)

$$\min_h \frac{1}{m_2} \sum_{j=1}^{m_2} (\tilde{y}_j - < h, \phi(\tilde{x}_j) \otimes \mu_{W|X=\tilde{x}_j, Z=\tilde{z}_j} >)^2 + \lambda_{BRG} ||h||_{\mathcal{H}}^2. \tag{10}$$

The optimal closed-form solution for the bridge can be obtained as (Mastouri et al., 2021)

$$\hat{h}_0 = \sum_{i=1}^{m_1} \sum_{j=1}^{m_2} \alpha_{i,j} \phi(\tilde{x}_j) \otimes \phi(w_i), \tag{11}$$

where $vec(\alpha) = (I \bar{\otimes} \Gamma)(\lambda_{BRG} m_2 I + \Sigma)^{-1} \tilde{Y}$ is the vectorized version of $\alpha$ and $\Sigma = (\Gamma^T \mathcal{K}_W \Gamma) \odot \mathcal{K}_{\tilde{X}}$ and $\Gamma = (\mathcal{K}_Z \odot \mathcal{K}_X + \lambda_{CME} m_1 I)(\mathcal{K}_{Z\tilde{Z}} \odot \mathcal{K}_{X\tilde{X}})$.

## 5.2 Acquisition function and sample selection

Given a pool of candidate samples without proxies $\mathcal{D}_{\mathcal{PL}}$. The goal is to select a set of samples from $\mathcal{D}_{\mathcal{PL}}$ that decreases predictive uncertainty (Smith et al., 2023). Define design $d = (x, z)$ and target design $d_* = (x_*, z_*) \sim q(d_*)$. In our setting, EPIG criterion becomes

$$EPIG(d) = \mathbb{E}_{q(d_*)}[I(y; y_*|d, d_*)]. \tag{12}$$

By considering that both CME and bridge are trained by ridge regression, which corresponds to a Gaussian process (Kanagawa et al., 2018), we conclude that the outputs also follow a joint Gaussian distribution. As a result, $[I(y; y_*|d, d_*)]$ has a closed form $I(y; y_*|d, d_*) = -\frac{1}{2}\log(1 - \rho^2(d, d_*))$ obtained based on conditional entropies, where $\rho(d, d_*) = \frac{Cov(y, y_*|d, d_*)}{\sqrt{\mathbb{V}(y|d)\mathbb{V}(y_*|d_*)}}$ (see Appendix B). Although based on this equation a closed-form for EPIG based on predictive variances ($\mathbb{V}$) and covariances ($Cov$) is achievable, evaluation of this equation is intractable in our setting due to the following reasons:

- The bridge function is maintained as a point estimate. As a result, the posterior uncertainty of the bridge and outcome level prediction variance are unavailable.

- At query time, the proxy $W$ is unobserved.

However, we use the CME posterior uncertainty as surrogate for EPIG, since the outcome is a linear functional of CME: $\hat{y} = T(d)(\mu_{W|x,z})$ where $T(d) = < h_{adp}, \phi(x) \otimes (.) >$. Since CMEs for each environment are trained on samples from that environment, we first filter samples by environment id $X^r = \{x_j^{(r)} : z_j^{(r)} = z\}$,

$Z^r = \{z_j^{(r)} : z_j^{(r)} = z\}$. The CME posterior uncertainty $\sigma_\mu$ (Chowdhury et al., 2020) is

$$\sigma_{\mu,r}^2(d) = \sigma_{\mu,r}^2(x,z) =$$
$$k(x,x)k(z,z) - (\Phi_{X^{(r-1)}}(x) \odot \Phi_{Z^{(r-1)}}(z))^\top (\mathcal{K}_{Z^{(r-1)}} \odot \mathcal{K}_{X^{(r-1)}} + n_{z,r-1}\lambda_{CME}I)^{-1}(\Phi_{X^{(r-1)}}(x) \odot \Phi_{Z^{(r-1)}}(z)),$$
(13)

where $n_{z,r-1}$ is the number of current queried samples at round $r-1$ with environment id $z$. Our acquisition selects samples from environments with large posterior uncertainty of the CME. In Appendix C, we formally show that PQAL with CME-based uncertainty acquisition can recover the maximal informative rank from candidate pool.

## 5.3 Bridge and CME functions adaptation

Consider a set of initial labeled samples from source domains ($\mathcal{D}_{\mathcal{LB}}$), and a set of queried samples with proxy at round $r$ ($\mathcal{D}_{\mathcal{PR}}^r$) from source and target environments. We leverage a small set of labeled target samples ($\mathcal{D}_{\mathcal{TLB}}^r$) to stabilize bridge adaptation in the finite-sample setting.

**Updating CMEs:** As CMEs for each environment are trained on samples from that environment, we first filter samples by environment id $\mathcal{D}_{\mathcal{PR}}^r(z) = \{(x_j^{(r)}, w_j^{(r)}) : z_j^{(r)} = z\}$ . We train CMEs for both source and target environments by adapting Equation 9 as

$$\mu_{W|X=x,Z=z}^{(r)} = \sum_{j \in \{j:z_j^{(r)}=z\}} b_j^{(r)}(x,z)\phi(w_j^{(r)}),$$
(14)

where $b^{(r)}(x,z) = (\mathcal{K}_{Z^{(r)}} \odot \mathcal{K}_{X^{(r)}} + \lambda_{CME}n_{r,z}I)^{-1}(\Phi_{Z^{(r)}}(z) \odot \Phi_{X^{(r)}}(x))$ and $|\mathcal{D}_{\mathcal{PR}}^r(z)| = n_{r,z}$. Also, $Z^{(r)}$ and $X^{(r)}$ are samples of $X$ and $Z$ in $\mathcal{D}_{PR}$ in iteration $r$.

**Adaptation of bridge:** After initial training of bridge function $\hat{h}_0$ on source domains (Equation 11), we obtain optimal parameters $\alpha_s$. We fine-tune this bridge function in the target domain using samples queried during training. $\hat{h}_0$ has a stable representation due to the abundance of covariate and proxy pairs in the source domains. However, during the active learning loop, only a few target samples are available, which risks overfitting and destabilizing the optimization. To prevent these issues, we represent target samples in the fixed feature space derived from the source domains. As a result, the bridge is adapted to the target domain while maintaining the stability established during source training. The predicted outcome $f_\alpha$ is defined over fixed source basis $\{\phi(w_i)\}_{i=1}^{m_1}$ and $\{\phi(\tilde{x}_j)\}_{j=1}^{m_2}$ as

$$f_\alpha(x,w) = \sum_{i=1}^{m_1} \sum_{j=1}^{m_2} \alpha_{i,j} k_\mathcal{W}(w_i,w).k_\mathcal{X}(\tilde{x}_j,x),$$
(15)

where $\alpha \in \mathbb{R}^{m_1 \times m_2}$ are the bridge parameters. At the start of adaptation, we initialize $\alpha^{(0)} := \alpha_s$. Then, we adapt a bridge by solving the regularized optimization problem

$$\hat{\alpha}^{(r)} = \arg\min_\alpha (\mathcal{L}_{source}(\alpha) + \lambda_{tgt}\mathcal{L}_{target}^{(r)}(\alpha) + \lambda_{sim}\mathcal{L}_{manifold}^{(r)}(\alpha) + \lambda_{reg} \| \alpha - \alpha^{(0)} \|_2^2).$$

$$\mathcal{L}_{source}(\alpha) = \frac{1}{|\mathcal{D}_{\mathcal{LB}}|} \sum_{i=1}^{|\mathcal{D}_{\mathcal{LB}}|} (y_i - f_\alpha(x_i,w_i))^2.$$

$$\mathcal{L}_{target}^{(r)}(\alpha) = \frac{1}{|\mathcal{D}_{\mathcal{TLB}}^r|} \sum_{j=1}^{|\mathcal{D}_{\mathcal{TLB}}^r|} (y_j^{(r)} - f_\alpha(x_j^{(r)},w_j^{(r)}))^2. \quad (16)$$

$$\mathcal{L}_{manifold}^{(r)}(\alpha) = \sum_{j=1}^{|\mathcal{D}_{\mathcal{TLB}}^r|} \sum_{k=1}^{|\mathcal{D}_{\mathcal{PR}}^r|} \mathcal{S}_{jk}(f_\alpha(x_j^{(r)},w_j^{(r)}) - f_\alpha(x_k^{(r)},w_k^{(r)}))^2.$$

In Equation 16, the first component $\mathcal{L}_{source}(\alpha)$ evaluates how well the current bridge, parameterized by $\alpha$, fits the source data. This loss prevents catastrophic forgetting (Kemker et al., 2018). The second

component $\mathcal{L}^{(r)}_{target}(\alpha)$ minimizes the prediction error on a few queried target samples with labels. The third component $\mathcal{L}^{(r)}_{manifold}(\alpha)$ encourages smooth prediction over the target manifold via the similarity kernel $\mathcal{S}$ between generated outcomes from labeled and unlabeled target samples. We set $\mathcal{S}$ to a Gaussian RBF kernel on the joint space $\mathcal{X} \times \mathcal{W}$. At prediction time, the final output for $(x_{new}, z_{new})$ is $\hat{y}_{new} = \langle h_{adp}, \phi(x_{new}) \otimes \mu_{W|x_{new}, z_{new}} \rangle$ and $h_{adp} = \sum_{i=1}^{n_1} \sum_{j=1}^{n_2} \alpha_{i,j} \phi(\tilde{x}_j) \otimes \phi(w_i)$.

## 6 Experiments

We evaluate our approach on synthetic regression tasks consistent with the causal DAG and assumptions in Section 2. After outlining the data generation process and baselines, we study the effect of imperfect proxies on identifiability, test the robustness of our method to varying degrees of distribution shifts, and evaluate the effectiveness of the proposed acquisition function. For experiment setup please refer to Appendix E.3.

### 6.1 Data generation process

Verifying the accuracy of our domain adaptation method requires known structural ground truth that complies with latent shift assumptions. To circumvent this, standard approach is to use semi-synthetic benchmarks (Louizos et al., 2017). By blending real-world covariates with synthetic confounding mechanisms, we can establish a reliable baseline to accurately measure performance. Following this methodology, we consider a diverse set of synthetic and semi-synthetic data generation processes, ranging from controlled regression tasks and the dSprites image dataset (Tsai et al., 2024) to complex, real-world tabular benchmarks, such as IHDP (Ruth et al., 2024) and ACS Folktables (Ding et al., 2021). Crucially, across all evaluated datasets, we explicitly design the generation process for $W$ to exhibit the characteristics of imperfect proxies.

**Dataset 1 (Continuous proxy):** We sample $U$ from a beta distribution with parameters $a$ and $b$. We generate a continuous proxy $W$ using a sinusoidal mapping with Gaussian noise $\epsilon$, where $B$ controls the degree of non-injectivity of the mapping from $U$ to $W$. Equation 17 shows a structural causal model (SCM) for Dataset 1.

**Dataset 2 (Discrete proxy):** We also consider the case of a discrete proxy, as some approaches, such as Prashant et al. (2025), rely on the discrete proxy assumption. In addition, we consider a nonlinear mapping from $U$ and $X$ to $Y$. Equation 18 shows a SCM for Dataset 2.

$$
\begin{aligned}
U &\sim Beta(a,b) \\
W = \sin(2\pi BU) + \epsilon \quad &\epsilon \sim \mathcal{N}(0, \sigma_w^2) \\
X &\sim \mathcal{N}(0,1) \\
Y &= (2U-1)X
\end{aligned}
\tag{17}
$$

$$
\begin{aligned}
U &\sim Beta(a,b) \\
W &= \begin{cases} \lfloor U.B \rfloor & \text{with probability } 1-\eta \\ R & \text{with probability } \eta \end{cases} \\
X &\sim \mathcal{N}(0,1) \\
Y &= U^3 X
\end{aligned}
\tag{18}
$$

Where $R$ is sampled uniformly from bins ($R \sim Unif\{0, 1, \ldots, B-1\}$) and $B$ is the number of bins to control imperfectness in the proxy. $\eta$ is the probability of proxy corruption. When the proxy is corrupted, W is reassigned uniformly at random to one of the $B$ bins.

**Dataset 3 (dSprites):** To demonstrate PQAL's ability in handling more complex datasets, we use the semi-synthetic dSprites dataset (Matthey et al., 2017). Data is generated according to the following SCM, where the latent confounder $U$ is the object orientation. Function $f_{image}$ generates an image from latent generative factors. For our experiments, we vary only orientation as a confounder, while holding all other generative factors (e.g., shape and scale) fixed. Figure A1 shows the effect of shifts in the distribution of $U$ in the generated image and the data is created as

$$
\begin{aligned}
U &\sim Beta(a,b) \\
W = \sin(2\pi BU) + \epsilon \quad &\epsilon \sim \mathcal{N}(0, \sigma_w^2) \\
X &\sim f_{image}(u) + \epsilon_x \\
Y &= (2U-1)X
\end{aligned}
\tag{19}
$$

**Dataset 4 (IHDP Benchmark):** We adapt the Infant Health and Development Program (IHDP) dataset (Ruth et al., 2024), which is built from a real-world clinical trial on premature infants. We partition the data by birth weight to construct distinct source and target environments ($Z$). We retain the actual clinical covariates ($X$) and environment assignments ($Z$) to preserve realistic marginal distributions. To establish an accurate baseline, we synthetically generate the unobserved confounder $U$, the outcome $Y$, and the imperfect proxy $W$. Intuitively, $U$ represents unmeasured clinical risks, such as parental stress or unrecorded genetic predispositions. The outcome $Y$ can be viewed as a clinical severity score indicating patient health status. Finally, $W$ can be interpreted as noisy risk indicator such as parental risk score.

$$U \sim Beta(a,b)$$
$$W = Scale(\sin(2\pi BU) + \epsilon_w) \quad \epsilon_w \sim \mathcal{N}(0, \sigma_w^2) \tag{20}$$
$$Y = (2U-1)X_{proj} + \epsilon_y \quad \epsilon_y \sim \mathcal{N}(0, \sigma_y^2)$$

where Scale(.) denotes standardization to zero mean and unit variance, $X_{proj}$ is the mean projection of observed covariates.

**Dataset 5 (ACS Folktables):** We utilize the Folktables (Ding et al., 2021) dataset that provides records containing highly entangled socioeconomic features such as education, age, and occupation across different states of the United States. We retain the actual demographic covariates $X$ and geographic environment assignment ($Z$) to preserve realistic marginal distributions. Because true confounders are inherently latent in this dataset, we establish an accurate baseline by synthetically generating the unobserved regional confounder $U$, the imperfect proxy $W$, and the outcome $y$. Intuitively, $U$ is an unmeasured regional socioeconomic pressures, such as the localized cost of living. Also, $Y$ can be viewed as an economic mobility score that summarizes overall financial success of individuals. Finally, $W$ can be interpreted as housing affordability score.

$$U \sim Beta(a,b)$$
$$W = Scale(\cos(2\pi BU) + \epsilon_w) \quad \epsilon_w \sim \mathcal{N}(0, \sigma_w^2) \tag{21}$$
$$Y = (2U-1)X_{proj} + \epsilon_y \quad \epsilon_y \sim \mathcal{N}(0, \sigma_y^2)$$

## 6.2 Effect of imperfect proxies on LECs

Here, we show how proxy imperfection affects the required diversity for environments to point-identify the predictor $\mathbb{E}[Y|X=x, Z=S+1]$, serving as the empirical diagnostic for the cross-domain rank condition and LEC proxy spanning (Assumption 3) discussed in Section 4.2. As explained in section 4.2, the conditional mean embedding for a fixed $x$ can be expressed as mixture over LECs: $\mu_{x,z} = \sum_{j=1}^{K} \pi_j(x,z)m_j$ where $m_j := \mathbb{E}[\phi(W)|U \in O_j]$. Since $m_j$s are invariant across environments, any observable variation between the conditional mean embeddings is based on mixing weights $\Pi(x)$. Therefore, identifying linear independence (environment variation) in mixture weights is mathematically equivalent to the evaluation of the singular values of a matrix that is formed by stacking CMEs of different environments. We report these values as the effective rank.

We use Dataset 2 for this experiment as discrete proxy bins provide a direct control over proxy non-injectivity. Figure 3 shows that as the number of bins increases (fine-grained proxies), the effective rank increases, suggesting environments become more distinguishable in CME space. In the limit with an infinite number of samples, this rank converges to the rank of the population stacked CME matrix (upper-bounded by the number of total environments; e.g. 3 for the case of 2 sources and 1 target).

## 6.3 Robustness under varying degrees of distribution shift

We examine the robustness of PQAL to varying degrees of shift alongside four closely related approaches. Proxy-DA (Tsai et al., 2024) is the closest approach to our study, and identifies the predictor $\mathbb{E}[Y|X=x, Z=S+1]$ but uses the same bridge in both the source and target environments and trains CMEs in the target environment based on given proxies. Prashant et al. (2025) (SC) train a mixture-of-experts (MOE) model to address latent distribution shift in discrete confounders. To adjust for the effects of confounders, the expert models are reweighted in the target using the recovered distribution of latent confounders inferred

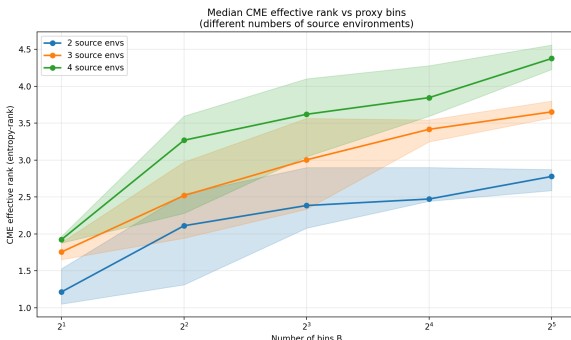

Figure 3: Effective rank with respect to proxy imperfectness for different number of environments

Table 1: MSE across all datasets. Blue and bold mark the lowest and second-lowest MSEs.

| Dataset1 | | | | | |
|---|---|---|---|---|---|
| Strategy | Degree 1 | Degree 2 | Degree 3 | Degree 4 | Degree 5 |
| FewShot_ERM | **0.0729** | 0.1708 | 0.3061 | 0.5229 | 0.7278 |
| Proxy_DA | 0.1083 | 0.2902 | 0.4194 | 0.8638 | 1.0926 |
| Oracle | 0.0473 | 0.0814 | 0.1093 | 0.1721 | 0.1974 |
| PQAL | 0.0800 | **0.1388** | **0.2025** | **0.3293** | **0.4015** |
| **Dataset2** | | | | | |
| FewShot_ERM | 0.0071 | 0.0212 | 0.0576 | 0.1594 | 0.2787 |
| Proxy_DA | 0.0109 | 0.0240 | 0.0732 | 0.1934 | 0.4112 |
| SC | 0.0098 | 0.0380 | 0.1013 | 0.2547 | 0.4727 |
| Oracle | 0.0037 | 0.0144 | 0.0339 | 0.0784 | 0.1101 |
| PQAL | **0.0053** | **0.0183** | **0.0466** | **0.1201** | **0.2216** |
| **dSprites** | | | | | |
| FewShot_ERM | 0.1869 | 0.604 | 0.969 | 1.199 | 1.256 |
| Proxy_DA | **0.054** | 0.334 | 0.708 | 1.004 | 1.071 |
| Oracle | 0.050 | 0.160 | 0.310 | 0.427 | 0.449 |
| PQAL | 0.055 | **0.303** | **0.641** | **0.888** | **0.946** |
| **IHDP** | | | | | |
| FewShot_ERM | 0.057 | 0.081 | 0.094 | 0.091 | 0.086 |
| Proxy_DA | **0.034** | 0.082 | 0.1 | 0.104 | 0.110 |
| Oracle | 0.032 | 0.053 | 0.064 | 0.061 | 0.066 |
| PQAL | 0.039 | **0.059** | **0.068** | **0.067** | **0.067** |
| **Folktables** | | | | | |
| FewShot_ERM | 0.038 | 0.068 | 0.101 | 0.137 | 0.158 |
| Proxy_DA | 0.034 | 0.062 | 0.089 | 0.121 | 0.150 |
| Oracle | 0.031 | 0.057 | 0.078 | 0.104 | 0.128 |
| PQAL | **0.032** | **0.058** | **0.082** | **0.109** | **0.132** |

from proxies. Furthermore, we compare our approach to few-shot-ERM, which minimizes risk using a few labels from the target sample, to show the necessity of adjusting for the effects of latent confounders. Finally, we also considered Oracle, which has access to all samples in the target pool, providing a lower bound for the errors. For a detailed explanation of baselines, please refer to Appendix E.1.

For Dataset 1, 3, 4, 5, we compare all methods except SC, which requires a discrete proxy. All baseline methods are evaluated using a random acquisition strategy, as they either do not use the CME in their structure or are not designed for active learning. PQAL actively selects samples based on CME uncertainty. To quantify varying degrees of distribution shift for Datasets 1-3, we keep the mean of the Beta distribution approximately fixed and vary its variance by selecting different parameters for the distribution. The variance of the Beta distribution is given by $\frac{ab}{(a+b+1)(a+b)^2}$. We use the resulting variances as discrete levels of distribution shift, with larger variance corresponding to stronger latent distribution shift. For IHDP and Folktables datasets (Datasets 4 and 5), we create a shift by progressively deviating the mean of the beta distribution. By monotonically adjusting the shape parameters $(\alpha, \beta)$, we shift the expected value of the unobserved confounder toward the extreme tail of the distribution. Therefore, PQAL performance is also evaluated under different directional shifts in the latent confounder mean. (Please refer to Appendix E.3 for details on the degree of shift and experiment setups).

Table 1 summarizes the mean square error (MSE) across these shift levels for all datasets. For the smallest degree of shift, Few-Shot ERM and Proxy-DA outperform our approach for dataset 1, IHDP and dSprite, respectively. However, for other shift degrees PQAL achieves lowest MSE. As the degree of shift increases, PQAL exhibits substantially better robustness than other methods, highlighting the importance of actively adapting to latent confounder shifts.

## 6.4 Effect of acquisition function

We evaluate the proposed acquisition function (CME uncertainty) against expected error reduction (EER)-inspired (Cohn et al., 1994) surrogate that selects representative $x$ (RPX). In addition, we compare our method with a random and an uncertainty-based acquisition function for $Z$ (refer to Appendix E.2 for a detailed explanation of each acquisition strategy). As discussed in section 5.2, defining any acquisition function based on the uncertainty of the bridge is intractable, and as a result, we do not include any bridge-based acquisition strategies in our experiments.

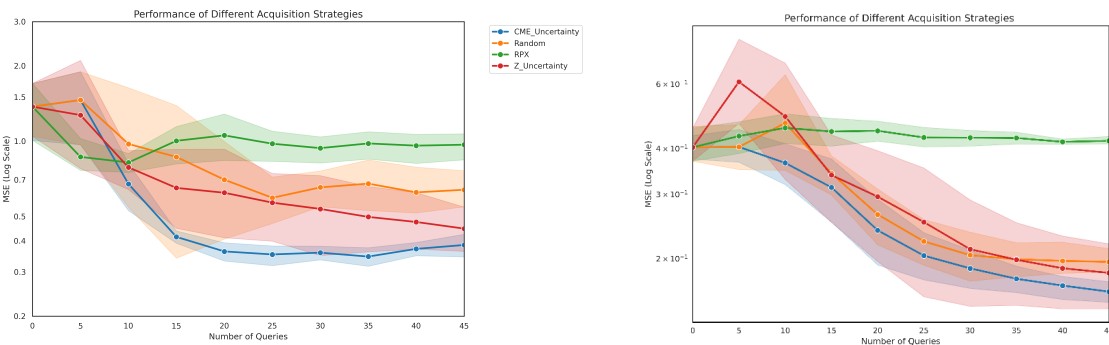

Figure 4: MSE error for different acquisition functions on Dataset 1(left) and Dataset 2 (right)

Figure 4 shows the MSE of PQAL using different acquisition functions for both synthetic datasets. The shaded region indicates the 95% confidence interval across random seeds. The CME uncertainty has the lowest MSE and the lowest variability compared to all baselines. Random acquisition improves with obtaining more queries, but remains worse than CME uncertainty in the continuous case. $Z$ uncertainty outperforms random acquisition by encouraging the model to cover environments. However, it performs worse than CME uncertainty as it does not account for the effect of covariates when selecting candidates to obtain a proxy. EER initially selects covariate representative points that reinforce stable, biased patterns. This selection leads to an increase in MSE as the model overfits the misspecified bridge function, which is blind to latent shifts.

## 7 Related work

**Robust predictor under distribution shift:** Several studies leverage proxies to train a robust predictor under latent distribution shifts. Alabdulmohsin et al. (2023) use concept and proxy variables to estimate latent subgroup density ratios, reweight conditional subgroup probabilities, and recover the predictor in the target environment without labels. Tsai et al. (2024) use PCI to adapt a trained model from the source to the target environment. They use the same bridge function in both source and target domains and update CMEs using target proxies. Prashant et al. (2025) recover the conditional latent confounder distributions based on proxies under a discrete latent confounder shift. They train a mixture-of-experts (MOE) model (Xu et al., 1994) for each value of latent confounders. To adapt the weights of experts in MOE for the target domain, they used recovered conditional latent confounder distributions. All aforementioned approaches assume some form of completeness to establish point-identification guarantees. In contrast, we establish point-identification guarantees for the predictor when completeness fails due to imperfect proxies, and recovering latent subgroups and latent variables is not possible. In addition, rather than assuming the diversity condition holds for environments, our framework actively seeks to satisfy it by querying the most informative samples. Furthermore, similar to Tsai et al. (2024), our approach applies to both discrete and continuous proxies and latent confounders.

**Causal identification in proxy-based causal inference:** In causal inference, the use of a proxy to adjust for the effect of unobserved confounders is well established. These approaches often rely on completeness for continuous and invertibility of $P(W \mid U)$ in discrete cases (Miao et al., 2018). Recent studies relax completeness at the expense of partial identification (set identification). Ghassami et al. (2023) obtain sharp bounds on the average treatment effect (ATE) by enforcing dependencies between proxies, treatment, and outcome. Zhang (2022) bound average causal effect (ACE) by assuming $P(W|U)$ is partially observable and bounded. While these approaches break completeness, their focus is on the set identification of causal effects such as ATE and ACE. Besides, a source of completeness violation, which is the effect of imperfect proxies, is not formalized in these studies.

## 8 Discussion

While our framework successfully relaxes the completeness assumption to provide point-identification guarantees under imperfect proxies, it relies on specific structural conditions. We outline the failure cases where LEC-based identification becomes too weak:

- **Violation of Remark 1 (Degenerate LEC)**: If the proxy is entirely uninformative about the latent confounder $U$, it fails to partition the latent space into at least two distinct classes ($|O| = 1$). In this degenerate case, the proxy provides no measurable information regarding domain-induced changes in $U$, making the domain adaptation problem unsolvable.

- **Violation of Definition 3 (Rank-Deficient Environment):** Even if the proxy successfully partitions the latent space ($|O| \geq 2$), LEC-based identification remains too weak if the observed source domains are homogeneous. If the environment does not mix the proxy-induced LECs in linearly independent ways, the cross-domain matrix $\Pi(X)$ becomes rank-deficient. Consequently, the target predictor falls back to being set-identified rather than point-identified.

- **Violation of Assumption 3 (Incompatibility)**: If outcome-relevant variations in the latent confounder fell entirely within a single LEC and thus cannot be distinguished by proxy, the target regression vector $r_x$ will not lie in the range of $\Pi(X)g(x)$. In this scenario, the LEC-level inverse problem is fundamentally unsolvable, and no valid bridge function exists.

- **Violation of Assumption 1 (Loss of invariance):** Our identification relies on the invariant causal mechanisms $P(Y \mid U, X)$ and $P(W \mid U)$ across domains. If the target domain introduces a structural shift in the physical data generation process (e.g., $P_{source}(Y \mid U, X) \neq P_{target}(Y \mid U, X)$), the bridge function learned in the source domain becomes misspecified.

## 9 Conclusion

This paper addresses domain adaptation under a latent shift with imperfect proxies. By introducing LECs to capture proxy-induced indistinguishability, we showed that point-identification of robust predictors is possible through cross-domain rank condition. We proposed PQAL to enforce this rank condition. Our experiments supported our claims in the successful adaptation of the predictor to the target environment.

## Acknowledgments

This work was supported by the Research Council of Finland (Flagship preprogram: Finnish Center for Artificial Intelligence FCAI and decision 3597/31/2023), ELLIS Finland, EU Horizon 2020 (European Network of AI Excellence Centers ELISE, grant agreement 951847; ERC ODD-ML 101201120; 3597/31/2023, grant agreement 3597/31/2023), UKRI Turing AI World-Leading Researcher Fellowship (EP/W002973/1). We also acknowledge the computational resources provided by the Aalto Science-IT Project from Computer Science IT.

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

# A    Proof of theorems

In this section we provides extended proof for Lemma 1 and Theorems .

## A.1    Proof of Lemma 1

*Proof.* Let $(\mathcal{A}, \mathcal{W})$ be any measurable set where $A \in \mathcal{A}$. By law of expected iteration (Durrett, 2019), we have:

$$P(W \in A | X = x, Z = z) = \mathbb{E}[1\{W \in A\} | X = x, Z = z] =$$
$$\mathbb{E}[\mathbb{E}[1\{W \in A\} | U, X = x, Z = z] \Big| X = x, Z = z] =$$
$$\mathbb{E}[P(W \in A | U, X = x, Z = z) \Big| X = x, Z = z] \tag{22}$$

Based on causal DAG 1, $W \perp (X, Z) \mid U$, hence:

$$P(W \in A | U, X = x, Z = z) = P(W \in A | U) \tag{23}$$

By replacing Equation 23 in Equation 22, we have

$$P(W \in A | X = x, Z = z) = \mathbb{E}[P(W \in A | U) \Big| X = x, Z = z] \tag{24}$$

Based on regularity assumption (Assumption 2), LECs $\{O_j\}_{j=1}^{|O|}$ form a measurable partition of the support of $U$ (Definition 2). Applying the law of total probability over this partition leads to:

$$P(W \in A | X = x, Z = z) = \mathbb{E}[P(W \in A|U) \big| X = x, Z = z] = \sum_{j=1}^{|O|} P(W \in A \mid U \in O_j) P(U \in O_j | X = x, Z = z) \tag{25}$$

Define $P_j(A) := P(W \in A | U \in O_j)$ and $\pi_j(x, z) := P(U \in O_j | X = x, Z = z)$, we have:

$$P(W \in A | X = x, Z = z) = \sum_{j=1}^{|O|} \pi_j(x, z) P_j(A) \tag{26}$$

$\square$

## A.2 Proof of Theorem 1

*Proof.* We show that the rank condition in Definition 3 is an LEC-level relaxation of the first completeness assumption. The first completeness assumption requires injectivity over all square-integrable functions of $U$, whereas the rank condition only requires injectivity over functions that are constant within each LEC.

Let $R \subset L^2(U)$ be the finite-dimensional subspace of functions that are constant on each LEC.

$$R = span\{1\{u \in O_j\} : j = 1, \dots, | O |\}. \tag{27}$$

Any $l \in R$ has the form:

$$l(u) = \sum_{j=1}^{|O|} \alpha_j 1\{u \in O_j\} \tag{28}$$

For such l, using the definition of LECs,

$$\mathbb{E}[l(U) \mid X = x, Z = z] = \sum_{j=1}^{|O|} \alpha_j P(U \in O_j \mid X = x, Z = z) = \sum_{j=1}^{|O|} \alpha_j \pi_j(x, z) \tag{29}$$

If completeness holds, then injectivity holds over all of $L^2(U)$. Since $R \subset L^2(U)$, completeness also implies injectivity on $R$. Therefore, completeness distinguishes all LEC-constant functions, but it also requires much more: it must distinguish functions that are within each LEC.

Next, we show that the rank condition also gives injectivity on $R$, without requiring full completeness. Given distinguishing environment set $Z^* = \{z_1, \dots, z_e\}$, if

$$\mathbb{E}[l(U) \mid X = x, Z = z_i] = 0 \tag{30}$$

for every $z_i \in Z^*$, then

$$\Pi(x)\alpha = 0, \tag{31}$$

where $\alpha = (\alpha_1, \dots, \alpha_{|O|})^T$. By Definition 3, $\Pi(x)$ has full rank $| O |$ for almost every $x$. Therefore, $\alpha = 0$, and hence $l(U) = 0$ almost surely for all $l \in R$. Thus, the rank condition ensures injectivity on the LEC-constant subspace $R$. However, the cross-domain rank condition does not imply completeness.

Assume proxies are imperfect, so there are some LEC $O_k$ that contains at least two latent values $\{u_1, u_2\}$. Suppose:

$$P(U = u_1 \mid X = x, Z = z) = P(U = u_2 \mid X = x, Z = z) \quad \text{for all } (x, z) \tag{32}$$

Define a function $l$ supported on $O_k$ by $l(u_1) = 1$, $l(u_2) = -1$ and $l(u) = 0$ for all $u \notin \{u_1, u_2\}$. Then $\mathbb{E}[l(U) \mid X = x, Z = z] = 1.P(u_1 \mid x, z) - 1.P(u_2 \mid x, z) = 0$ for all $(x, z)$. Thus, completeness fails. However, the aggregated LEC mixture weight is:

$$\pi_k(x, z) = P(U \in O_k \mid X = x, Z = z) = P(u_1 \mid x, z) + P(u_2 \mid x, z) \tag{33}$$

which can be linearly independent from the mixture weights of other LECs. Thus, the LEC rank condition holds while completeness fails. Therefore, the cross-domain rank condition is weaker from a LEC-level perspective: it only requires separation across LECs, while completeness requires separation over the full latent space, including within-LEC variation.

$\square$

## A.3 Proof of Theorem 2

*Proof.* **Existence:** Fix a distinguishing environment set $Z^\star$ (Definition 3) and let $r(x, z) = \mathbb{E}[Y \mid X = x, Z = z]$. By Lemma 1, for each $z \in Z^\star$ and almost surely $x$,

$$p(w \mid x, z) = \sum_{j=1}^{K} \pi_j(x, z) p_j(w), \quad K = \mid O \mid . \tag{34}$$

For any measurable function $h(x, w)$, the conditional expectation operator satisfies:

$$(\mathcal{T}_z h)(x) = \int_{\mathcal{W}} h(x, w) p(w \mid x, z) dw = \sum_{j=1}^{K} \pi_j(x, z) \int_{\mathcal{W}} h(x, w) p_j(w) \tag{35}$$

Define LEC moment functions:

$$g_j(x) := \int_{\mathcal{W}} h(x, w) p_j(w) dw, \quad j = 1, \ldots, K, \tag{36}$$

and write $(g_1(x), \ldots, g_k(x))^\top$.

For fixed $x$, stacking Equation 35 over $z_l \in Z^\star$ to obtain the linear system:

$$r_x = \Pi(x) g(x), \tag{37}$$

where $r_x := (r(x, z_l))_{l=1}^{e} \in \mathbb{R}^e$ and $\Pi(x) \in \mathbb{R}^{e \times K}$ has entries $\Pi_{lj}(x) = \pi_j(x, z_l)$.

By the compatibility condition in Assumption 3, for almost surely $x$ there exists a measurable $g(x) \in \mathbb{R}^K$ satisfying Equation 37. Moreover, by Assumption 3.3 $g_j \in L_2(\mathcal{X})$ for all $j$. Next, by Assumption 3.2, $p_j \in L_2(\mathcal{W})$ and gram matrix

$$G \in \mathbb{R}^{K \times K}, \quad G_{jk} := \int_{\mathcal{W}} p_j(w) p_k(w) dw, \tag{38}$$

is invertible. Define the coefficient vector $\alpha(x) := G^{-1} g(x)$ and construct

$$h_0(x, w) := \sum_{j=1}^{K} \alpha_j(x) p_j(w), \tag{39}$$

we first verify that $h_0 \in L_2(\mathcal{X} \times \mathcal{W})$. Since $g \in L_2(\mathcal{X})^K$ and $a(x) = G^{-1}g(x)$, we have $a_j \in L_2(\mathcal{X})$ for all $j$. In addition, each $p_j \in L_2(\mathcal{W})$ and $K$ is finite. To show $h_0 \in L_2(\mathcal{X} \times \mathcal{W})$:

$$
\begin{aligned}
\| h_0 \|^2_{L_2(\mathcal{X} \times \mathcal{W})} &= \int_{\mathcal{X}} \int_{\mathcal{W}} | \sum_{j=1}^{K} a_j(x)p_j(w)|^2 dw dx \leq \\
K \sum_{j=1}^{K} \int_{\mathcal{X}} \int_{\mathcal{W}} |a_j(x)|^2 |p_j(w)|^2 dw dx &= K \sum_{j=1}^{K} ||a_j(x)||^2_{L_2(\mathcal{X})} ||p_j(w)||^2_{L_2(\mathcal{W})} < \infty.
\end{aligned}
\tag{40}
$$

where the first inequality is derived from Cauchy-Schwarz theorem (Young, 1988) and the second equality is application of Fubini's theorem (Durrett, 2019).

Now we compute the moments of $h_0$. For each $k \in \{1, \ldots, K\}$,

$$
\int_{\mathcal{W}} h_0(x, w)p_k(w) dw = \sum_{j=1}^{K} a_j(x) \int_{\mathcal{W}} p_j(w)p_k(w) dw = (Ga(x))_k = g_k(x)
\tag{41}
$$

By replacing $h$ in Equation 35 with $h_0$ in Equation 41, for each $z_l \in Z^\star$ we have:

$$
(\mathcal{T}_{z_l} h_0)(x) = \sum_{j=1}^{K} \pi_j(x, z_l)g_j(x) = (\Pi(x)g(x))_l = r(x, z_l),
\tag{42}
$$

where the last equality follows from Equation 37. Hence, $\mathcal{T}_z h_0 = r(., z)$ for all $z \in Z^\star$. Therefore,

$$
\mathcal{T}_{joint} h = r
\tag{43}
$$

As a result, at least one bridge function exist.

**Set identification:** Let $h_1$ and $h_2$ be two solutions of $\mathcal{T}_{joint} h = r$. Then

$$
\mathcal{T}_{joint} h_1 = r = \mathcal{T}_{joint} h_2 \Rightarrow \mathcal{T}_{joint}(h_2 - h_1) = 0
\tag{44}
$$

So $h_2 - h_1 \in ker(\mathcal{T}_{joint})$. Conversely, for any let $d \in ker(\mathcal{T}_{joint})$, we have

$$
\mathcal{T}_{joint}(h_0 + d) = r
\tag{45}
$$

Thus the solution set is:

$$
\mathcal{H}(r) = \{h : \mathcal{T}_z h = r(., z)(\forall z \in Z^\star)\} = h_0 + ker(\mathcal{T}_{joint}) = h_0 + \bigcap_{z \in Z^\star} ker(\mathcal{T}_z)
\tag{46}
$$

$\square$

## A.4 Proof of Theorem 3

*Proof.* By Theorem 2, $\mathcal{T}_{joint} h = r$ has at least one solution in $\mathcal{H}(r) = \{h : \mathcal{T}_{joint} h = r\}$. Based on $\mathcal{T}_{joint} h = (\mathcal{T}_z h)_{z \in Z^\star}$, $\mathcal{T}_{joint} h = r$ is equivalent to

$$
\mathcal{T}_z h(.) = r(., z) \qquad z \in Z^\star
\tag{47}
$$

To show point identification of predictor, given any $h_1, h_2 \in \mathcal{H}_r$, define $\Delta h = h_2 - h_1$. Then

$$\mathcal{T}_{joint}\Delta h = \mathcal{T}_{joint}h_1 - \mathcal{T}_{joint}h_2 = r - r = 0 \tag{48}$$

Therefore, $\Delta h \in ker(\mathcal{T}_{joint})$. By definition of $\mathcal{T}_{joint}$, this is equivalent to:

$$\mathcal{T}_z \Delta h = 0 \qquad \forall z \in Z^\star \tag{49}$$

By Lemma 1, the conditional distribution proxy $W$ has a mixture representation.

$$P(W|X = x, Z = z) = \sum_{j=1}^{|O|} \pi_j(x, z) P(W|U \in O_j) \tag{50}$$

Recall the definition of the conditional expectation operator $\mathcal{T}_z$ acting on any function h:

$$(\mathcal{T}_z h)(x) = \int_w h(x, w) P(w|x, z) dw \tag{51}$$

Substituting the mixture representation from Lemma 1 into this integral:

$$(\mathcal{T}_z h)(x) = \int_{\mathcal{W}} h(x, w) \Big( \sum_{j=1}^{|O|} \pi_j(x, z) P(w|U \in O_j) \Big) dw \tag{52}$$

By the linearity of the integral, we can exchange the sum and integral:

$$(\mathcal{T}_z h)(x) = \sum_{j=1}^{|O|} \pi_j(x, z) \underbrace{\int_{\mathcal{W}} h(x, w) P(w|U \in O_j) dw}_{\mathcal{T}_{O_j} h} \tag{53}$$

Thus, the operator itself decomposes as:

$$\mathcal{T}_z = \sum_{j=1}^{|O|} \pi_j(x, z) \mathcal{T}_{O_j} \tag{54}$$

where $\mathcal{T}_{O_j}$ is the operator defined by the conditional distribution of the proxy given the $j$-th equivalent classes. Similarly based on invariance of causal structure (Assumption 1) we have:

$$\mathcal{T}_{S+1} = \sum_{j=1}^{|O|} \pi_j(x, S+1) \mathcal{T}_{O_j}. \tag{55}$$

By Definition 3 for almost every $x$, the matrix of distinguishing environment set $\Pi(x) = [\pi_j(x, z_l)]_{l,j}, l = 1, \ldots, e, j = 1, \ldots, |O|$ has full rank $K = |O|$. Since dimension of mixture space is $K$, the vectors $\{\pi(x, z) : z \in Z^\star\}$ span entire space $\mathbb{R}^K$. As a result, the mixture weight vector $\pi(x, S+1) \in \mathbb{R}^K$ lies in the linear span of source mixture weights. Therefore, there is coefficient $\{\gamma_z(x)\}_{z \in Z^\star}$ such that:

$$\mathcal{T}_{S+1} = \sum_{z \in Z^\star} \gamma_z(x) \mathcal{T}_z \tag{56}$$

Following the same steps leading to Equations 48 and 49 for target domain we have:

$$\mathcal{T}_{S+1}\Delta h = \sum_{z \in Z^\star} \left(\gamma_z(x)\mathcal{T}_z\right)\Delta h = \sum_{z \in Z^\star} \gamma_z(x)\underbrace{\left(\mathcal{T}_z\Delta h\right)}_{0} = 0 \tag{57}$$

This implies $\mathcal{T}_{S+1}h_1 = \mathcal{T}_{S+1}h_2$. Thus, the predictor $\mathbb{E}[Y|X, Z = S+1]$ is uniquely identified. □

## B Derivation of acquisition function

Both CME and bridge training are performed via ridge regression, which corresponds to a Gaussian process (Kanagawa et al., 2018). As a result, the output also follows a joint Gaussian distribution. Therefore, we can replace the information gain $[I(y; y_*|d, d_*)]$ with its closed form $[I(y; y_*|d, d_*)] = \mathbb{H}(y|d) + \mathbb{H}(y_*|d_*) - \mathbb{H}(y, y_*|d, d^*)$ where $\mathbb{H}$ is entropy. Each entropy has a closed form as follows:

$$\mathbb{H}(y|d) = \frac{1}{2}\log(2\pi e\mathbb{V}(y|d))$$

$$\mathbb{H}(y_*|d_*) = \frac{1}{2}\log(2\pi e\mathbb{V}(y_*|d_*)) \tag{58}$$

$$\mathbb{H}(y; y_*|d, d_*) = \frac{1}{2}\log((2\pi e)^2 det(\Sigma))$$

Where $\Sigma = \begin{pmatrix} \mathbb{V}(y|d) & Cov(y, y_*|d, d_*) \\ Cov(y, y_*|d, d_*) & \mathbb{V}(y_*|d_*) \end{pmatrix}$ and $\mathbb{V}$ is variance and $Cov$ is covariance matrix. Then, mutual information can be obtained by:

$$I(y; y_*|d, d_*) = [\frac{1}{2}\log(2\pi e\mathbb{V}(y|d))] + [\frac{1}{2}\log(2\pi e\mathbb{V}(y_*|d_*))] - [\frac{1}{2}\log((2\pi e)^2 det(\Sigma))] =$$
$$\frac{1}{2}\log[\frac{\mathbb{V}(y|d)\mathbb{V}(y_*|d_*)}{det(\Sigma)}] = \frac{1}{2}\log[\frac{\mathbb{V}(y|d)\mathbb{V}(y_*|d_*)}{\mathbb{V}(y|d)\mathbb{V}(y_*|d_*) - Cov(y, y_*|d, d_*)}] \tag{59}$$

Define $\rho(d, d_*) = \frac{Cov(y, y_*|d, d_*)}{\sqrt{\mathbb{V}(y|d)\mathbb{V}(y_*|d_*)}}$ and therefore $Cov(y, y_*|d, d_*)^2 = \rho(d, d_*)^2\mathbb{V}(y|d)\mathbb{V}(y_*|d_*)$. By substituting $Cov(y, y_*|d, d_*)^2$ in Equation 59 we get:

$$I(y; y_*|d, d_*) = \frac{1}{2}\log[\frac{\mathbb{V}(y|d)\mathbb{V}(y_*|d_*)}{\mathbb{V}(y|d)\mathbb{V}(y_*|d_*) - \rho^2(d, d_*)\mathbb{V}(y|d)\mathbb{V}(y_*|d_*)}] \tag{60}$$

Then, by canceling out similar terms, we have:

$$I(y; y_*|d, d_*) = -\frac{1}{2}\log(1 - \rho^2(d, d_*)) \tag{61}$$

## C Guarantees for CME uncertainty acquisition

In this section, we show that the acquisition function based on CME uncertainty recovers resolvable rank.

Based on (Kanagawa et al., 2018), in the case when observations are noise-free and the Gram matrix $\mathcal{K}_{XZ}$ is invertible , the standard deviation of posterior $(\sigma_\mu(x, z) = \sqrt{k(x, x)k(z, z) - (\Phi_X(x) \odot \Phi_Z(z))^\top(\mathcal{K}_{XZ})^{-1}(\Phi_X(x) \odot \Phi_Z(z)))}$, will be equal to worst case error $sup_{f \in \mathcal{H}_k, ||f||_{\mathcal{H}_k} \leq 1}\left(\sum_{i=1}^n \omega_i(x, z)f(x_i, z_i) - f(x, z)\right)$ where $(\omega_1(x, z), \ldots, \omega_n(x, z)) = \mathcal{K}_{XZ}^{-1}(\Phi_X(x) \odot \Phi_Z(z))$. In CME setting, the goal of ridge regression is learning $\mathbb{E}[g(W) \mid X = x, Z = z] = < g, \mu_{x,z} >$ so the worst case error becomes $sup_{||g||_{\mathcal{H}_W} \leq 1} |< g, \mu_{x,z} > - \sum_{i=1}^n \omega_i(x, z) < g, \mu_{x_i,z_i} >| = sup_{||g||_{\mathcal{H}_W} \leq 1} | \mu_{x,z} - \sum_{i=1}^n \omega_i(x, z)\mu_{x_i,z_i} |$. Intuitively, this means whenever the posterior variance is zero, the conditional

mean embeddings $\mu_{x,z}$ lie in the linear span of previously observed embeddings and introduce no new independent direction. Whereas a positive posterior variance certifies the presence of a new linearly independent component. In practice, observed data contains noise, and a regularization term $\lambda_{CME}$ is used when $\mathcal{K}_{XZ}$ is not invertible. However, in such conditions, we can assume a relaxed version of this condition holds by considering an error bound.

**Assumption 4** (Approximate variance span correspondence). *For a fixed $x \in \mathcal{X}$, let $\mathcal{M}_{r-1} := span\{\mu_{x,z_1}, \ldots, \mu_{x,z_{r-1}}\} \subset \mathcal{H}_{\mathcal{W}}$ be the span of CMEs selected up to round $r-1$. Assume there exist constants $\epsilon \geq 0$ and $\eta > 0$ such that for all rounds $r$ and and all candidates $z \in \mathcal{Z}_{\mathcal{PL}}$:*

$$\mu_{x,z} \in \mathcal{M}_{r-1} \Rightarrow \sigma_\mu^2(x, z) \leq \varepsilon$$
$$dist(\mu_{x,z}, \mathcal{M}_{r-1}) \geq \eta \Rightarrow \sigma_\mu^2(x, z) \geq \varepsilon + \eta \tag{62}$$

*Here $dist(\mu, \mathcal{M}) := inf_{m \in \mathcal{M}} \| \mu - m \|_{\mathcal{H}_{\mathcal{W}}}$ denotes the RKHS distance to subspace $\mathcal{M}$.*

**Remark 3** (Regularization constraint). *We note that satisfying Assumption 4 implicitly constrains the regularization parameter $\lambda_{CME}$ in CME posterior uncertainty in Equation 13. Specifically, $\lambda_{CME}$ must be sufficiently small such that it does not dampen the posterior variance of orthogonal components below the separation margin $\varepsilon + \eta$. In our experiments, we set $\lambda_{CME}$ as a hyperparameter and tune it to maintain this sensitivity.*

While the identification results allow an arbitrary set of environments, in practice, the learner can only select from a finite set of available environments. We therefore analyze a pool-based policy that selects z from the candidate pool $\mathcal{Z}_{\mathcal{PL}}$. The following theorem shows that by using CME uncertainty, the resolvable rank within this pool is recoverable, which is the finite-sample analogue of the distinguishing set rank used for identification.

**Definition 4** (Resolvable pool rank). *Let $\zeta_\eta(x)$ be largest integer $\zeta$ for which there exist environments $z_1, \ldots, z_\zeta \in \mathcal{Z}_{\mathcal{PL}}$ such that for each $j = 1, \ldots, \zeta$,*

$$dist(\mu_{x,z_j}, span\{\mu_{x,z_1}, \ldots, \mu_{x,z_{j-1}}\}) \geq \eta \tag{63}$$

*Equivalently $\zeta_\eta(x)$ is the maximum number of linearly independent directions in $\mathcal{H}_{\mathcal{W}}$ that are separable from each other by a margin $\eta$.*

**Theorem 4** (PQAL acquisition and resovable rank). *For fixed $x \in \mathcal{X}$, define data pool information rank*

$$rank_{\mathcal{PL}}(x) := dim(span\{\mu_{x,z} : z \in \mathcal{Z}_{\mathcal{PL}}\}) < \infty \tag{64}$$

*Consider Assumption 4 holds. Consider the acquisition policy:*

$$z_r \in arg\ max_{z \in \mathcal{Z}_{\mathcal{PL}}} \sigma_\mu^2(x, z). \tag{65}$$

*Then the selected CMEs satisfy*

$$dim(span\{\mu_{x,z_1}, \ldots, \mu_{x,z_r}\}) \geq \min\{r, \zeta_\eta(x)\}. \tag{66}$$

*In particular, after $\zeta_\eta(x)$ rounds, the selected set covers all CME directions in the pool that are resolvable at scale $\eta$.*

*Proof.* Let $\mathcal{M}_{r-1} := span\{\mu_{x,z_1}, \ldots, \mu_{x,z_{r-1}}\}$. If $dim(\mathcal{M}_{r-1}) < \zeta_\eta(x)$, then by definition of $\zeta_\eta(x)$ there exist some $\bar{z} \in \mathcal{Z}_{\mathcal{PL}}$ such that:

$$dist(\mu_{x,\bar{z}}, \mathcal{M}_{r-1}) \geq \eta \tag{67}$$

By Assumption 4, this implies $\sigma_\mu^2(x, \bar{z}) \geq \varepsilon + \eta$. Since $z_r$ maximizes $\sigma_\mu^2(x, .)$, we have $\sigma_\mu^2(x, z_r) \geq \varepsilon + \eta$. On the other hand, if $\mu_{x,z_r} \in \mathcal{M}_{r-1}$, Assumption 4 would imply $\sigma_\mu^2(x, z_r) \leq \varepsilon$, contradicting $\sigma_\mu^2(x, z_r) \geq \varepsilon + \eta$. Therefore $\mu_{x,z_r} \notin \mathcal{M}_{r-1}$, and hence $dim(\mathcal{M}_r) = dim(\mathcal{M}_{r-1}) + 1$. If instead $dim(\mathcal{M}_{r-1}) \geq \zeta_\eta(x)$, the lower bound $dim(\mathcal{M}_r) \geq \zeta_\eta(x)$ is immediate. The claim follows by induction.

$\square$

## D  Algorithm

---

**Algorithm 1** Proximal Quasi Bayesian Active learning (PQAL)

---

**Require:** $\mathcal{D}_{\mathcal{LB}} = \{(x_i, w_i, z_i, y_i)\}_{i=1}^m$: Data set with labels from source domains, $\mathcal{D}_{\mathcal{PL}} = \{(x_i, z_i,)\}_{i=1}^n$: Pool of data samples without label and proxy Budget $R$, Empty set of samples with proxy $\mathcal{D}_{\mathcal{PR}}$, Empty set of samples with proxy and label $\mathcal{D}_{\mathcal{TLB}}$, number of proxy queries per round $v_p$, number of labeled queries per round $v_{lb}$

**Ensure:** Bridge function parameter $\alpha$, CMEs $\{\mu_{W|x,z}\}_{z \in \mathcal{Z}}$, And optimal predictor $\mathbb{E}[Y|X = x, Z = z]$

1: **Initial training:**
2: Train CMEs for each environment by using closed form of Equation 9
3: Train bridge by using its closed form 11 and obtain bridge parameter $\alpha_0$
4: **while** Budget $r \leq R$ **do**
5:     **if** r=1 **then**
6:         Select $v_p$ and $v_{lb}$ samples randomly
7:     **else**
8:         Select $v_p$ and $v_{lb}$ samples samples based on acquisition function in Equation 13
9:         $((x, z) \in \arg\max_{(x,z)} = \sigma_{\mu,r}^2(x, z))$
10:     **end if**
11:     Query $w$ from external source of information for $v_p$ selected samples $D_{\mathcal{PR}}^r = D_{\mathcal{PR}}^r \cup \{(x_i, z_i, w_i)\}_{i=1}^{v_p}$
12:     Query $w$ , $y$ from external source of information for $v_{lb}$ selected samples
13:     $D_{\mathcal{TLB}}^r = D_{\mathcal{TLB}}^r \cup \{(x_i, z_i, w_i, y_i)\}_{i=1}^{v_{lb}}$
14:     Remove $v_p$ and $v_{lb}$ samples from the pool of candidate samples
15:     $D_{\mathcal{PL}} = D_{\mathcal{PL}} \setminus \{(x, z) : (x, z) \in (D_{\mathcal{PR}}^r \cup D_{\mathcal{TLB}}^r)\}$
16:     Train CME $\mu_{W|X=x,Z=z}^{(r)}$ in target domain by using equation 14
17:     Adapt bridge function parameter $\alpha^{(r)}$ by Equation 16
18: **end while**
19: **return** $\alpha^{(r)}, \{\mu_{W|x,z}^{(r)}\}_{z \in \mathcal{Z}}$

---

## E  Experiment details

In this section we introduce each baseline with details.

### E.1  Baselines

- **Proxy-DA**: Tsai et al. (2024) assume completeness and prove the trained bridge in source domain can be used on target domains ($h_s = h_t$). As a result, only updating CMEs in target domain is enough to adapt the predictor and get robust predictor.

- **SC**: Prashant et al. (2025) uses discrete proxies $W$ to capture the structure of unobserved variables. This approach first trains an encoder to learn the latent distribution $P(U|X)$ by using $(X, W)$. A mixture of experts is then trained on source data, where predictions are combined by using inferred latent at inference time, the model estimates the target latent marginal by comparing the distribution of latent components for the queried target covariates against the source. These estimates are used to recalibrate the mixture weights, allowing the model to adapt its predictions to target domain shift.

- **Few-shot-ERM:** In few-shot Empirical risk minimization (ERM), we concatenate data from source and target domains and train a MLP over concatenation of current source samples and few queried target samples $\{(x, y)\}_{(x,y) \in \mathcal{D}_{\mathcal{LB}} \cup \mathcal{D}_{\mathcal{T}_{\mathcal{LB}}}}$ and predictor is obtained by minimizing loss over these samples.

- **Oracle:** Oracle baseline has access to all target samples from candidate pool. It is once trained on source labeled samples plus all labeled samples from candidate pools. We then fine tune shred source bridge based on loss source and target losses in Equation 16 ($\hat{\alpha}^{(r)} = \arg\min_\alpha (\mathcal{L}_{source}(\alpha) + \lambda_{tgt}\mathcal{L}_{target}^{(r)}(\alpha) + \lambda_{reg} \parallel \alpha - \alpha^{(0)} \parallel_2^2$).

## E.2 Compared Acquisition strategies

- **Representative $x$ (RPX):** We use an expected error reduction (EER) inspired surrogate that favors representative samples in covariate space. Concretely, given the current unlabeled target pool $\{x_i\}_{i=1}^n$, we form the kernel gram matrix $\mathcal{K}_X \in \mathbb{R}^{n \times n}$. Each candidate $x_i$ is scored by its (regularized) global influence,

$$s_i = \frac{\sum_{j=1}^n k^2(x_i, x_j)}{k(x_i, x_i) + \lambda} \tag{68}$$

  where $\lambda > 0$ is a ridge parameter. Intuitively nominator $\sum_{j=1}^n k^2(x_i, x_j)$ measures how strongly $x_i$ is correlated with the remainder of the pool in kernel space. Thus, points with high scores are central and capture the structure that is shared by many samples in the candidate pool. Selecting these points tends to a reduction in the aggregated posterior variance and a tractable approximation for EER.

- **Z-uncertainty (environment coverage acquisition):** This strategy prioritizes coverage of underrepresented environments. At each active learning round, we compute the empirical frequency of each environment value $z$ among current labeled samples. Then we assign each candidate $(x_i, z_i)$ a score that is inversely proportional to the number of labeled points that are already collected from the same environment as candidate $s_i = \frac{1}{n_{z_i}+1}$, where $n_{z_i}$ denotes the current labeled count for environment $z_i$. This acquisition favors environments that have been observed less.

- **Random:** This strategy randomly selects a point from the candidate pool.

## E.3 Experiment setup

For all experiments, we use 35 samples from each source environment at the beginning of the active learning loop and query 5 samples at each iteration. For PQAL and Few-shot-ERM, 2 of the 5 samples are equipped with label $y$ and proxy $w$, and the remaining 3 receive only proxy $w$. In the following, we mention the specific settings for each experiment. For the robustness and acquisition experiments, we set $B = 4$.

1. **Effect of imperfect proxies on LECs:** We ran experiments with varying numbers of source environments $(2, 3, 4)$ and a total budget of 45 queries across 6 different seeds. We use following beta distributions for source distributions $\{(2, 10), (4, 8), (8, 4), (10, 2)\}$ we fix target distributions to $(6, 6)$.

2. **Robustness under varying degrees of distribution shift on synthetic datasets:** We ran experiments with 2 source environments and a total budget of 60 queries across 4 different seeds. We use following beta distributions for source distributions $\{(2, 4), (2.1, 3.9)\}$, we vary degrees of shift in target distributions by using $\{(8.0, 12.0), (6.0, 6.0), (5.0, 3.333), (3.0, 1.286), (2.0, 0.5)\}$ beta distributions.

3. **Robustness under varying degrees of distribution shift on dSprites dataset:** We ran experiments with 2 source environments and a total budget of 60 queries across 6 different seeds. We use following beta distributions for source distributions $\{(2, 4), (2.1, 3.9)\}$, we vary degrees of shift in target distributions by using $\{(10, 10), (15, 5), (25, 3), (40, 1), (60, 0.5)\}$ beta distributions. Figure A1 shows the shift in image orientation from source to target environment.

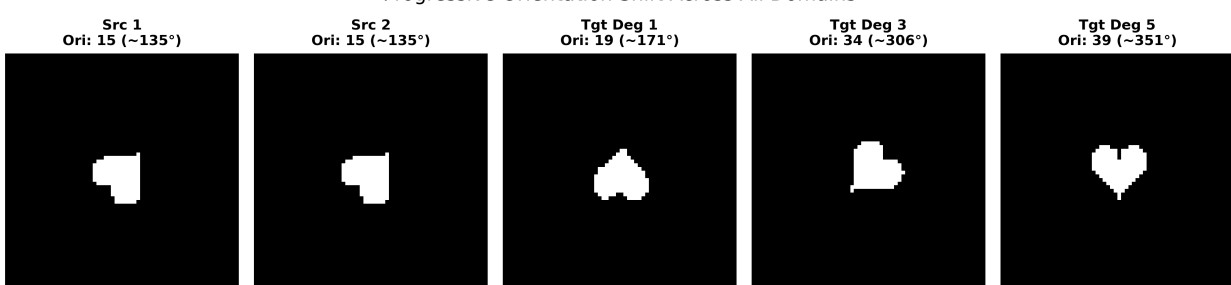

Figure A1: Change in latent variable orientation from source to target domains

4. Robustness under varying degrees of distribution shift on IHDP dataset: We ran experiments with 2 source environments and a total budget of 60 queries across 6 different seeds. We partition the data into the domain $Z$ based on birth weight to simulate distinct clinical environments. source 1 consists of infants with normal/high birth weight, source 2 includes Medium/low birth weight infants, and the target domain is comprised of extremely low birth weight infants. The latent confounder $U$ is conditioned on these weight-based partitions to model distinct underlying risk profiles. We use the beta distributions $\{(2,5),(5,2)\}$ for the source environments. We quantify the degree of shift by varying the directional mean of target distributions using parameters $\{(3,4),(1,5,5),(1.33,5.17),(1.17,5.33),(1,5.5)\}$ which progressively shifted the expected value of $U$ toward the extreme tail to model unobserved clinical risks.

5. Robustness under varying degrees of distribution shift on Folktables dataset: We conducted experiments using 2 source environments (California and Texas) and a total budget of 60 queries across 6 different seeds. We partition the data into domain $Z$ based on US states to simulate distinct regional socioeconomic environments, conditioning the latent confounder $U$ on these state-level assignments. We use the beta distributions $\{(2,5),(5,2)\}$ for the source environments. We quantify the degree of shift by varying the directional mean of target distributions using parameters $\{(3,4),(1,5,5),(1,5.5),(0.5,6),(0.2,6.5)\}$ for states Nevada, New York, Pennsylvania, Ohio, and Alabama, respectively. As a result, the expected value of $U$ is shifted toward the extreme tail to model unobserved regional socioeconomic pressures.

6. **Effect of acquisition function:** We ran experiments with 2 source environments and a total budget of 45 queries across 4 different seeds. We use following beta distributions for source distributions $\{(2,4),(2.1,3.9)\}$, and a target distribution $\{(2.0,0.5)\}$.

Note that the source and target distributions in the first experiment is differ from those in the the second and third experiments, as we wanted to maximize latent heterogeneity to ensure any reduction in rank is due to proxy imperfection rather than environment similarity. Conversely, for the robustness and acquisition experiments, we selected closer source environments, thereby isolating the challenge of adapting to significant distribution shifts in the target.

