# OpenReview forum: "Point-Identification of a Robust Predictor Under Latent Shift with Imperfect Proxies"
_TMLR — Accepted by TMLR_

### Review · Reviewer_VLke · 2026-04-15

**Summary Of Contributions:**

The paper studies domain adaptation under latent confounder shift when only imperfect proxy variables are available.
It introduces latent equivalent classes, which group latent confounder values that induce the same conditional proxy distribution, and argues that the target predictor can still be point-identified if the observed domains provide sufficient diversity in how they mix these classes.
Based on this idea, the paper proposes a cross-domain rank condition as an alternative to standard completeness assumptions and develops an active learning method, Proximal Quasi-Bayesian Active learning (PQAL), to query informative proxy labels.
Experiments on synthetic and semi-synthetic datasets are used to support the identification story and the proposed acquisition strategy.

**Audience:**

Yes

**Audience Explanation:**

The paper addresses proxy-based domain adaptation under latent shift, which is a relevant problem for researchers working on causal ML, distribution shift, and identification under hidden confounding.
The main idea may interest part of the TMLR audience, even though I am not fully convinced by the current technical presentation.

**Claims And Evidence:**

No

**Claims Explanation:**

- The central point-identification claim is supported only under substantially stronger assumptions than the narrative suggests.
In particular, Theorems 2 and 3 rely not just on the cross-domain rank condition, but also on Assumption 3, whose compatibility and spanning requirements appear to do essential work.
- The claim that the proposed rank condition is "strictly weaker than completeness" is not fully convincing from the proof as written.
The argument in Appendix A.2 is intuitive, but the key implication from completeness to existence of a full-rank distinguishing environment set is not established with enough care for such a central claim.
- The exposition also conflates distinct notions such as "invertibility of $P(W \mid U)$", rank conditions, and injectivity/completeness of conditional operators.
This weakens the conceptual clarity of the paper and makes it difficult to tell exactly which property is required for each result.

**Requested Changes:**

Clarify the exact assumptions needed for the main identification result and foreground the role of Assumption 3.
As written, the narrative suggests that the cross-domain rank condition is the key replacement for completeness, but the existence and point-identification results also rely on substantial compatibility and spanning assumptions that should be stated more prominently.

---

> ### Author Response · Authors · 2026-05-06
>
> We appreciate that the reviewer finds the problem relevant.
>
> **1-The central point-identification claim and role of Assumption 3**
>
> We agree that the exposition should more clearly separate the role of the cross-domain rank condition from the role of Assumption 3. Our claim is **not** that the cross-domain rank condition alone replaces all assumptions needed for point-identification of the target predictor. Rather, the rank condition relaxes the completeness/injectivity assumption, while Assumption 3 provides the solvability conditions needed for the existence of a valid bridge. In the revised paper, we added a paragraph in Section 3.1 after Equation 1 explaining the roles of regularity, solvability, and completeness assumptions. In addition, after Assumption 3, we added Remark 2 in Section 4.2 to foreground Assumption 3. In this remark, we explain that the compatibility, LEC proxy spanning, and square-integrability conditions are analogous to solvability conditions required for bridge existence in prior studies. Theorems 2 and 3 rely on both components: Assumption 3 for bridge existence/solvability and the cross-domain rank condition for proving the target predictor is point-identified once we establish that at least one valid bridge exists.
>
> **2-Strictly weaker claim and proof**
>
> We thank the reviewer for their careful evaluation of Theorem 1 and Appendix A.2. The reviewer correctly noted that the original proof did not establish that completeness implies the existence of a fixed full-rank distinguishing environment set. This was stronger than our intended claim. Our intended claim is that the cross-domain rank condition is a strict relaxation of completeness in the LEC-level sense: it only requires injectivity over functions that are constant within each proxy-induced LEC, while completeness requires injectivity over all square-integrable latent functions, i.e., over $L^2(U)$. In the revised version, we refined the statement of  Theorem 1 and revised Appendix A.2 to make this distinction explicit. The revised proof first
> establishes the LEC-level injectivity guaranteed by the rank condition and then shows strictness by constructing a case where completeness fails due to within-LEC indistinguishability while the aggregate LEC mixture weights still satisfy the rank condition.
>
> **3-Clarifying completeness, invertibility, and the cross-domain rank condition**
>
> We thank the reviewer for pointing out that the original exposition could more clearly distinguish between invertibility of $P(W|U)$, completeness/injectivity of conditional expectation operators, and our cross-domain rank condition. In the revised Section 3.1, we added a paragraph explicitly separating these notions.  In the discrete setting, the first completeness condition corresponds to the invertibility of the matrix representation of $P(U|X,Z)$, which captures how latent variables vary across covariates and environments. The second completeness condition corresponds to invertibility of the matrix representation of $P(W|U)$, which captures how informative the proxy is about the latent confounder. Our proposed condition is distinct from both: it is a cross-domain rank condition based on variation in latent distributions across environments, and is introduced to handle the case where the proxy mechanism is imperfect and the second completeness condition fails.

---

### Review · Reviewer_9om9 · 2026-04-20

**Summary Of Contributions:**

The paper studies domain adaptation under latent shift when available proxies for the latent confounder are imperfect, so standard completeness-based identification may fail. It introduces latent equivalent classes (LECs), shows that the target robust predictor can still be point-identified under a cross-domain rank condition that is claimed to be weaker than completeness, and proposes an active-learning method, PQAL, to query informative proxies and adapt the predictor. Empirically, PQAL is evaluated on two synthetic settings and a semi-synthetic dSprites benchmark, where it usually improves over baselines, especially at larger shift levels.

Strengths:
- novel problem formulation via LECs
- meaningful combination of theory and active learning
- empirical results are promising under stronger shifts.

Weaknesses:
- The theoretical assumptions remain fairly strong and somewhat abstract, especially the distinguishing-environment rank condition and Assumption 3
- The empirical study is limited to synthetic/semi-synthetic settings, and some claims around sample efficiency and minimal querying are not fully validated beyond these settings.

**Audience:**

Yes

**Audience Explanation:**

Yes. I expect this to be of interest to readers working on domain adaptation, causal/proxy-based learning, robustness under hidden confounding, and active learning.

**Broader Impact Concerns:**

I do not see a major immediate ethical concern.

**Claims And Evidence:**

Yes

**Claims Explanation:**

Partially yes. The main theoretical claims are clearly stated through the LEC decomposition, the weaker-than-completeness claim, and the point-identification theorem, with proofs deferred to the appendix. The empirical section also provides supportive evidence that PQAL is competitive and often best under larger latent shifts, and that the CME-uncertainty acquisition performs best among the tested acquisition rules.

That said, the evidence is not fully convincing for all claims. In particular, the theory depends on nontrivial assumptions such as compatibility, LEC proxy spanning, and sufficient cross-domain diversity, and it is not entirely clear how realistic or checkable these are in practice. Also, the experiments are confined to controlled settings, so the practical generality of the identification and active querying claims remains somewhat uncertain.

**Requested Changes:**

Major:
- better clarify and discuss the practical meaning, necessity, and verifiability of the main assumptions, especially the distinguishing-environment rank condition and Assumption 3.
- better delineate what is formally proven versus what is only suggested empirically, particularly regarding sample efficiency and “minimal” proxy querying.

Minor:
- expand discussion of limitations and failure cases, ideally including when LEC-based identification may still be too weak in practice.
- improve empirical breadth, for example, with a more realistic benchmark or stronger ablations on the number of environments, proxy quality, and queried target labels.

---

> ### Author Response · Authors · 2026-05-06
>
> We appreciate that the reviewer finds our formulation novel, the combination of proposed theory and practice meaningful, and our empirical results promising.
>
> **1-Clarification and practical perspective about Assumption 3 and the cross-domain rank condition**
>
> We agree with the reviewer that Assumption 3 and the cross-domain rank condition should also be clarified from a practical perspective and the roles of the different assumptions must be discerned. We address these concerns through the following modifications in the revised version. We added a paragraph in Section 3.1 after Equation 1  explaining the roles of regularity, solvability, and completeness assumptions. Next, by adding a paragraph after Theorem 1  in Section 4.2, we explain the practical meaning and verifiability of the cross-domain rank condition. In addition, after Assumption 3 we added Remark 2, which specifically discusses the necessity, practical meaning, and verifiability of each part of Assumption 3.
>
> **2-Clarification on sample efficiency and “minimal” proxy querying**
>
> We agree with the reviewer that the terms "minimal" and "sample efficiency" can imply a formal mathematical lower bound on sample complexity. Because our theoretical guarantees in Section 4 concern point identification, while our claims regarding the sample efficiency of PQAL are empirical as shown in Section 6, we have updated the text and removed  occurrences of minimal and sample efficiency and replaced them with terms such as "a small, targeted set", "strategically selecting these imperfect  proxy queries under strict budget constraints", "considering a limited budget", in the abstract, third paragraph of the introduction, and first paragraph of Section 5, respectively.
>
>
> **3-Limitations and failure cases of LEC-based identification**
>
> We agree with the reviewer that adding limitations and failure cases for our approach provides valuable context. In the revised version we added a Discussion section (Section 8) where we assess violations of important assumptions,  remarks, and  definitions that may occur in practice.
>
> **4-Improve empirical breadth**
> We agree with the reviewer on this point and added two more realistic benchmarks, including IHDP and Folktables datasets. The revised version includes modifications to the first paragraph of Section 6.1 and two new paragraphs at the end of Section 6.1 for introducing these new datasets.  We also extended Table 1 to include new experimental results. We  added a new paragraph in Section 6.3 to explain experiment details briefly. Additionally, we added two new paragraphs to Appendix E.3 to include details of the experimental setup of these two new benchmarks.

---

### Review · Reviewer_xxMk · 2026-04-22

**Summary Of Contributions:**

This paper studies domain adaptation when the proxy is imperfect so that completeness assumption does not hold. The authors shows that the predictor is point identifiable even though the bridge function is partially (set) identified. I have the following comments.

1. In Section 3.1, the authors compared with proximal causal inference (PCI) as motivation, and mentioned the completeness PCI imposed. This seems wired because the variables, data structure, and parameters to be identified in this paper are different from those in PCI. In PCI, $Z$ is a proxy that needs to be informative about the unobserved confounder, so completeness is relevant there. In this paper, $Z$ is the domain indicator.

2. Related, even in PCI, they do not need the bridge function to be uniquely identified; they only assume that there exists a bridge, then the target functional is identified. In this paper, Theorem 2 also allows the bridge to be set-identified, and Assumption 3 seems to add extra solvability conditions such as “Compatibility”.

3. The paper discusses completeness in terms of $P(W\mid U)$. But my understanding is how informative a proxy $W$ is about $U$ should be captured by the completeness in terms of $P(U|W)$. This seems nuanced and should be clarified.

4. The notation $E$ for an operator looks too close to expectation.

5. It would help to use clearer notation to clarify that the target of identification is $E[Y\mid X, Z=S+1]$, not a general $E[Y\mid X, Z]$.

**Audience:**

Yes

**Audience Explanation:**

Data adaptation under distribution shift is an important problem, and studying the assumptions under which such transfer is possible is meaningful.

**Claims And Evidence:**

No

**Claims Explanation:**

additional clarification is needed, especially to better justify the comparison to PCI.

**Requested Changes:**

See comments above.

---

> ### Author Response · Authors · 2026-05-06
>
> We appreciate that the reviewer finds the problem relevant and studying these assumptions meaningful. Regarding comments:
>
>
> **1-Connection to PCI and role of Z**
>
> Our paper **does not** use the standard PCI setup unchanged, **nor** do we compare against traditional PCI methods. Rather, we adapt the multi-domain predictor identification formulation of [1], which uses PCI-inspired identification techniques for domain adaptation. We already state this in the paper. For example, in section 3, we write: "Following Tsai et al. (2024), we   adapt  identification techniques  from  proximal causal inference (PCI) to provide point-identification guarantees for predictors." This distinction is important because our goal is not causal identification in traditional PCI sense. We are **not identifying causal effect** such as average treatment effect (ATE), and we do **not present** our result as
> **causal identification**. Rather, we use PCI techniques as tools to identify robust predictor under latent shift.
>
> In this formulation, Z is the domain/environment indicator, while W is the proxy of latent confounder U. This is the notation used in Tsai et.al (2024), and we preserve it because our contribution is to adress a limitation of that predictor-identification formulation under imperfect proxies and latent shift. Thus, in our paper, Z is not proxy variable;n Wis the proxy, and Z indexes environments with different latent distributions through P(U|Z=z).
>
> **2- Assuming existence of bridge and extra solvability assumptions**
>
> Bridge existence is not assumed in rigorous identification arguments; it relies on solvability conditions that guarantee such a bridge exists [1,2]. We **do not**  simply assume bridge existence as a primitive because, in our setting with imperfect proxies, the conditional expectation operator is inherently non-injective. In standard PCI with perfect proxies,  bridge existence may be  assumed directly, but this can hide the  underlying solvability and regularity conditions needed for a solution to exist. When proxies lose information, bridge existence requires specific structural alignment which we make explicit through Assumptions 2 and 3.  Assumption 2 provides regularity/measurability conditions. Assumption 3 provides the LEC-level solvability conditions needed to guarantee at least one bridge solution. Thus, Assumption 3 is analogous to standard solvability/range conditions rather than an extra assumption. In the revised  paper,  we clarified these points in Section 3.1 after Equation 1 by explaining the role of regularity, solvability, and completeness assumptions. We also revised Section 4.2 to draw a parallel between solvability assumptions in prior work and Assumption 3 in our paper.
>
> **3-Completeness in terms of P(W|U)**
>
> We formulate proxy informativeness in terms of the forward proxy mechanism $P(W|U)$, rather than the posterior $P(U|W)$. This is the standard formulation in proxy-based and proximal causal inference settings, where completeness is stated through conditional expectation operators induced by the proxy mechanism [1,2]. This distinction is important in our setting because, under latent shift, the distribution of U, and more generally P(U|X,Z), varies across environments. Therefore, P(U|W) depends both on the stable proxy mechanism P(W|U) and environment-specific latent distribution. Using P(U|W) would mix intrinsic proxy informativeness with domain-specific latent shift, whereas P(W|U) isolates the stable measurement mechanism from the latent confounder to the proxy.
>
> **4-Regarding notation E**
>
> We agree with the reviewer and we changed the notation to $\mathcal{T}$.
>
> **5-Regarding notation E[Y|X=x,Z=S+1]**
>
> The problem formulation already states that the goal of identification is $\mathbb{E}[Y|X=x,Z=S+1]$.  To avoid any possible ambiguity, we revised the remaining shorthand occurences of  $\mathbb{E}[Y|X,Z]$ in the theoritical and experimental sections to explicitly refer to target-domain predictor.
>
> **References**
>
> [1] Miao, Wang, Zhi Geng, and Eric J. Tchetgen Tchetgen. "Identifying causal effects with proxy variables of an unmeasured confounder." Biometrika 105.4 (2018): 987-993.
>
> [2] Tsai, Katherine, et al. "Proxy methods for domain adaptation." International Conference on Artificial Intelligence and Statistics. PMLR, 2024.

---

### Decision · Action_Editor_XVBs · 2026-05-27

**Recommendation:** Accept as is

**Audience:**

Yes

**Audience Explanation:**

Researchers in domain adaptation, distribution shift, causal inference would find this work of interest.

**Claims And Evidence:**

Yes

**Claims Explanation:**

This paper studies the domain adaptation problem under latent shift where the completeness assumption may not be satisfied.
To overcome the difficulty, the authors introduce the concept of latent equivalent classes (LECs), which are defined as groups of latent confounders that induce the same conditional proxy distribution.
The authors move forward and show that point-identification for the robust predictor remains achievable as long as multiple domains differ sufficiently in how they mix proxy-induced LECs to form the robust predictor.
This domain diversity condition is substantially weaker than completeness.
Further, the author introduce the Proximal Quasi-Bayesian Active learning (PQAL) framework, which actively queries a minimal set of diverse domains that satisfy this rank condition.
The authors conduct both synthetic data and semi-synthetic data analyses to show that,
PQAL can efficiently recover the point-identified predictor, demonstrating robustness to varying degrees of shift and outperforming previous methods.